# Continuous attractors for dynamic memories

**Davide Spalla[1]\*, Isabel Maria Cornacchia[1,2], Alessandro Treves[1]**

[1]SISSA – Cognitive Neuroscience, Via Bonomea, Trieste, Italy; [2]University of Turin – Physics Department, Torino, Italy

**Abstract** Episodic memory has a dynamic nature: when we recall past episodes, we retrieve not only their content, but also their temporal structure. The phenomenon of replay, in the hippocampus of mammals, offers a remarkable example of this temporal dynamics. However, most quantitative models of memory treat memories as static configurations, neglecting the temporal unfolding of the retrieval process. Here, we introduce a continuous attractor network model with a memory-dependent asymmetric component in the synaptic connectivity, which spontaneously breaks the equilibrium of the memory configurations and produces dynamic retrieval. The detailed analysis of the model with analytical calculations and numerical simulations shows that it can robustly retrieve multiple dynamical memories, and that this feature is largely independent of the details of its implementation. By calculating the storage capacity, we show that the dynamic component does not impair memory capacity, and can even enhance it in certain regimes.

## Introduction

The temporal unfolding of an event is an essential component of episodic memory. When we recall past events, or we imagine future ones, we do not produce static images but temporally structured movies, a phenomenon that has been referred to as 'mental time travel' (*Eichenbaum and Cohen, 2004*; *Tulving, 2002*).

The study of the neural activity of the hippocampus, known for its first-hand involvement in episodic memory, has provided many insights on the neural basis of memory retrieval and its temporal dynamics. An interesting example is the phenomenon of hippocampal replay, that is the reactivation, on a compressed time scale, of sequences of cells active in previous behavioral sessions. Replay takes place during sharp wave ripples, fast oscillations of the hippocampal local field potential that are particularly abundant during sleep and restful wakefulness (*Buzsáki et al., 1983*; *Buzsáki et al., 1992*). Indeed, replay has been observed during sleep (*Skaggs and McNaughton, 1996*; *Nádasdy et al., 1999*), inter-trial rest periods (*Foster and Wilson, 2006*; *Jackson et al., 2006*), and during still periods in navigational tasks (*Dupret et al., 2010*; *Pfeiffer and Foster, 2013*). Replay activity has been hypothesized to be crucial for memory consolidation (*O'Neill et al., 2010*) and retrieval (*Karlsson and Frank, 2009*), as well as for route planning (*Pfeiffer and Foster, 2013*; *Ólafsdóttir et al., 2018*).

A temporally structured activation takes place also before the exposure to an environment (*Dragoi and Tonegawa, 2011*), a phenomenon known as *preplay*, and a recent study showed that this dynamical feature emerges very early during development, preceding the appearance of theta rhythm (*Farooq and Dragoi, 2019*) in the hippocampus. The fact that hippocampal sequences are present before the exposure to the environment suggests that their dynamical nature is not specific to a role in spatial cognition, but is inherent to hippocampal operation in general. Moreover, in a recent study *Stella et al., 2019* have shown that the retrieved sequences of positions during slow wave sleep are not always replaying experienced trajectories, but are compatible with a random walk on the low dimensional manifold that represents the previously explored environment. This

**\*For correspondence:**
dspalla@sissa.it

**Competing interests:** The authors declare that no competing interests exist.

**eLife digest** When we recall a past experience, accessing what is known as an 'episodic memory', it usually does not appear as a still image or a snapshot of what occurred. Instead, our memories tend to be dynamic: we remember how a sequence of events unfolded, and when we do this, we often re-experience at least part of that same sequence. If the memory includes physical movement, the sequence combines space and time to remember a trajectory. For example, a mouse might remember how it went down a hole and found cheese there.

However, mathematical models of how past experiences are stored in our brains and retrieved when we remember them have so far focused on snapshot memories. 'Attractor network models' are one type of mathematical model that neuroscientists use to represent how neurons communicate with each other to store memories. These models can provide insights into how circuits of neurons, for example those in the hippocampus (a part of the brain crucial for memory), may have evolved to remember the past, but so far they have only focused on how single moments, rather than sequences of events, are represented by populations of neurons.

Spalla et al. found a way to extend these models, so they could analyse how networks of neurons can store and retrieve dynamic memories. These memories are represented in the brain as 'continuous attractors', which can be thought of as arrows that attract mental trajectories first to the arrow itself, and once on the arrow, to the arrowhead. Each recalled event elicits the next one on the arrow, as the mental trajectory advances towards the arrowhead. Spalla et al. determined that memory networks in the hippocampus of mammals can store large numbers of these 'arrows', up to the same amount of 'snapshot' memories predicted to be stored with similar models.

Spalla et al.'s results may allow researchers to better understand memory storage and recall, since they allow for the modelling of complex and realistic aspects of episodic memories. This could provide insights into processes such as why our minds wander, as well as having implications for the study of how neurons physically interact with each other to transmit information.

suggests that what is essential are not the sequences themselves, but the tendency to produce them: neural activity tends to move, constrained to abstract low-dimensional manifolds which can then be recycled to represent spatial environments, and possibly non-spatial ones as well. This dynamic nature extends to multiple timescales, as suggested by the observation that the neural map of the same environment progressively changes its component cells over time (*Ziv et al., 2013*).

Low-dimensional, dynamic activity is not constrained to a single subspace: replay in sleep can reflect multiple environments (*Dragoi and Tonegawa, 2013*; *Gridchyn et al., 2020*), the content of awake replay reflects both the current and previous environments (*Karlsson and Frank, 2009*), and during behavior fast hippocampal sequences appear to switch between possible future trajectories (*Kay et al., 2020*). Further evidence comes from a recent study with human participants learning novel word pair associations (*Vaz et al., 2020*). The study shows that the same pair-dependent neural sequences are played during the encoding and the retrieval phase.

A similar phenomenon – a dynamic activity on low dimensional manifolds – is present in memory schemata, cognitive frameworks that constrain and organize our mental activity (*Ghosh and Gilboa, 2014*), and have been shown to have a representation in the medial temporal lobe (*Baraduc et al., 2019*). Yet another example of dynamical, continuous memories is offered by motor programs, which have been described as low-dimensional, temporally structured neural trajectories (*Shenoy et al., 2013*; *Gallego et al., 2017*; *Oztop and Arbib, 2002*), or as dynamical flows on manifolds (*Huys et al., 2014*; *Pillai and Jirsa, 2017*).

We refer to these objects as *dynamical continuous attractors*, since they involve a continuous subspace that constrains and attracts the neural activity, and a dynamical evolution in this subspace. *Figure 1* schematically illustrates the concept of dynamical continuous attractors and their possible role in some of the neural processes described above.

In most cases, computational analyses of low-dimensional neural dynamics are not concerned with memory, and focus on the description of the features of single attractors, more than on their possible coexistence. On the other hand, mechanistic models of memory usually neglect dynamical aspects, treating memories as static objects, either discrete (*Amit et al., 1985*) or continuous

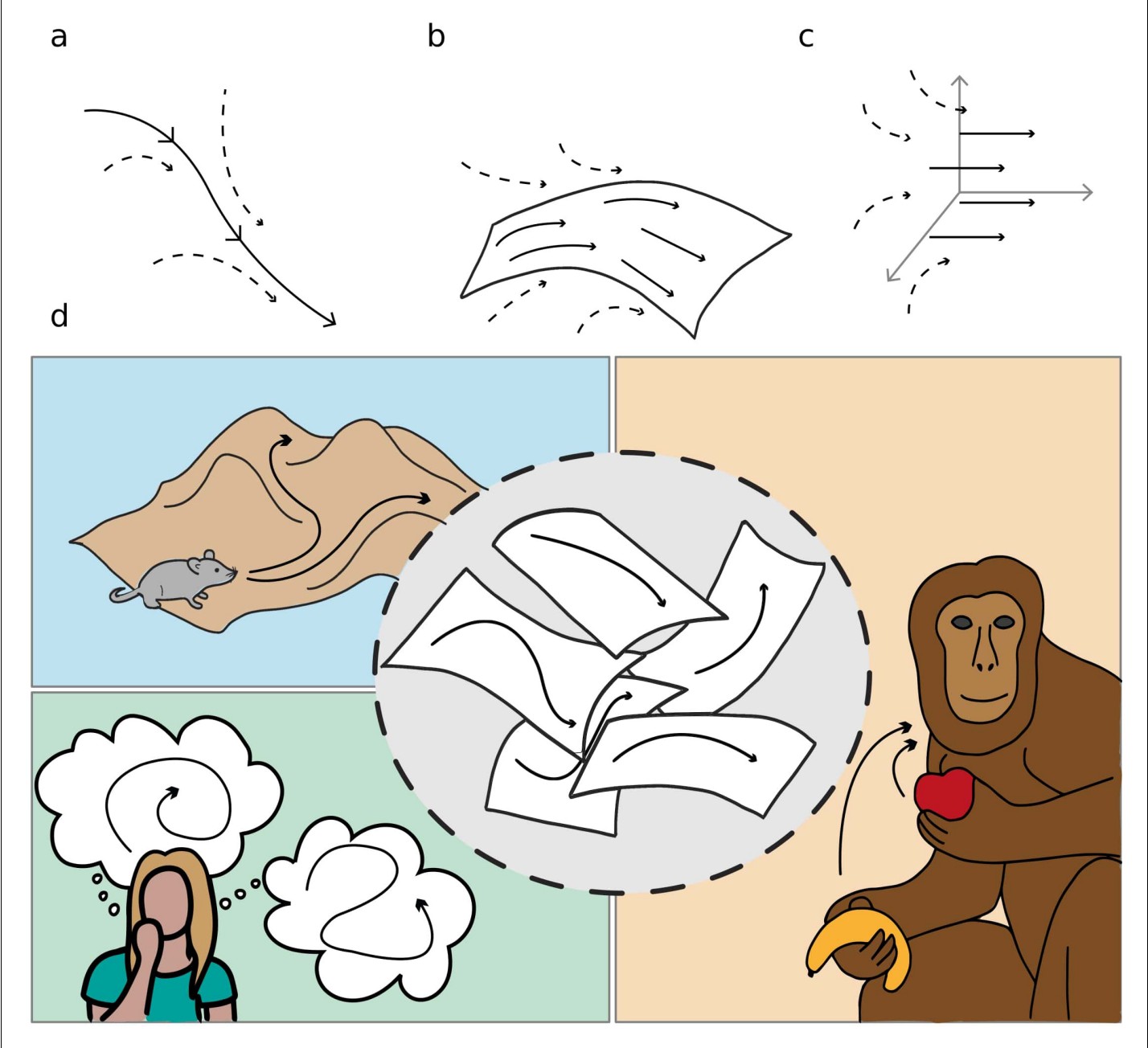

**Figure 1.** Schematic illustration of dynamic continuous attractors as a basis of different neural processes. Top row: a scheme of continuous attractive manifolds, with a dynamic component in 1D (a), 2D (b) and 3D (c). The neural activity quickly converges on the attractive manifold (dotted arrows), then slides along it (full arrows), producing a dynamics that is temporally structured and constrained to a low dimensional subspace. Bottom row: multiple dynamic memories could be useful for route planning (top left), involved in mind wandering activity (bottom left) or represent multiple learned motor programs (right).

(*Battaglia and Treves, 1998*; *Monasson and Rosay, 2013*; *Samsonovich and McNaughton, 1997*; *Spalla et al., 2019*). The production of sequences of discrete memories can be implemented with a heteroassociative component (*Sompolinsky and Kanter, 1986*), usually dependent on the time integral of the instantaneous activity, that brings the network out of equilibrium and to the next step in the sequence. A similar effect can be obtained with an adaptation mechanism in a coarse grained model of cortical networks (*Kropff and Treves, 2005*), with the difference that in this case the transitions are not imposed, but driven by the correlations between the memories in so-called latching

dynamics (*Russo et al., 2008*; *Kang et al., 2017*). Moreover, adaptation-based mechanisms have been used to model the production of random sequences on continuous manifolds (*Azizi et al., 2013*), and shown to be crucial in determining the balance between retrieval and prediction in a network describing CA3-CA1 interactions (*Treves, 2004*). In the case of continuous attractor networks, movement can be induced also by mechanisms that integrate an external velocity input and make use of asymmetric synaptic strengths. Models of this kind have been used for the description of head direction cells (*Zhang, 1996*), spatial view cells (*Stringer et al., 2005*), and grid cells (*Fuhs and Touretzky, 2006*; *Burak and Fiete, 2009*), and can represent simultaneously the positions of multiple features and their temporal evolution (*Stringer et al., 2004*). In the simplest instantiation, these systems do not necessarily reflect long-term memory storage: the activity is constrained on a single attractive manifold, which could well be experience independent.

Here, we propose a network model able to store and retrieve multiple independent dynamic continuous attractors. The model relies on a map-dependent asymmetric component in the connectivity that produces a robust shift of the activity on the retrieved attractive manifold. This connectivity profile is conceived to be the result of a learning phase in which the mechanism of spike timing dependent plasticity (STDP) (*Markram et al., 1997*) produces the asymmetry. Crucially, the asymmetry is not treated here as a 'pathological' feature, assumed to level out in the limit of long learning, but as a defining trait of the stored memories. The balance between two components – one symmetric and trajectory-averaged, the other asymmetric and trajectory-dependent – is explicit in the formulation of the model, and allows to study their effects on memory storage.

In what follows we develop an analytical framework that allows to derive the dependence of important features of the dynamics, such as the replay speed and the asymmetry of the activity cluster, as a function of the relevant parameters of the model. We show with numerical simulations that the behavior of the model is robust with respect to its details, and depends weakly on the shape of the interactions. Finally, we estimate the storage capacity for dynamical memories and we find it to be of the same order of the capacity for static continuous attractors, and even higher in some regimes.

## Modeling framework

### A mechanistic model for dynamic retrieval

The model we consider is a continuous attractor neural network, with an additional anti-symmetric component in the connectivity strength. We consider a population of $N$ neurons, with recurrent connectivity described by an interaction matrix $J_{ij}$, whose entries represent the strength of the interaction between neuron $i$ and $j$. The activity of each neuron is described by a positive real number $V_i \in \mathbb{R}^+$ representing its instantaneous firing rate. The dynamic evolution of the network is regulated by the equations:

$$\tau \frac{\partial V_i}{\partial t} + V_i = g\left[\left(\sum_{j \neq i} J_{ij} V_j - h_0\right)\right]^+ \tag{1}$$

where $[...]^+$ is the threshold linear activation function

$$[x]^+ = x\theta(x) \tag{2}$$

with the gain $g$ modulating the slope and the Heaviside step function $\theta(...)$ setting to zero sub-threshold inputs. The first term on the right hand side of *Equation 1* represent the excitatory inputs provided to neuron $i$ from the rest of the network through recurrent connections. The threshold $h_0$ and the gain $g$ are global parameters that regulate the average activity and the sparsity of the activity pattern (*Treves, 1990*).

In numerical simulations, these parameters are dynamically adjusted at each time step to constrain the network to operate at a certain average activity (usually fixed to one without loss of generality) and at a certain sparsity $f$, defined as the fraction of active neuron at each time (see appendix A). The connectivity matrix $J$ of the network encodes a map of a continuous parameter $\vec{x}$ spanning a low-dimensional manifold, e.g. the position in an environment. To do so, each neuron is assigned a preferential firing location $\vec{x}_i$ in the manifold to encode, and the strength of the interaction between

pairs of neuron is given by a decreasing, symmetric function of the distance between their preferred firing locations

$$J_{ij} \sim K_S(|\vec{x}_i - \vec{x}_j|). \tag{3}$$

This shape of the interactions is a typical one in the framework of continuous attractor neural networks (*Samsonovich and McNaughton, 1997*; *Battaglia and Treves, 1998*; *Tsodyks, 1999*) and is thought to come from a time-averaged Hebbian plasticity rule: neurons with nearby firing fields will fire concurrently and strengthen their connections, while firing fields that are far apart will produce weak interactions. The symmetry of the function $K_S$, usually called interaction kernel, ensures that the network reaches a static equilibrium, where the activity of the neurons represents a certain position in the manifold and, if not pushed, remains still.

## The shift mechanism

The assumption of symmetric interactions neglects any temporal structure in the learning phase. In case of learning a spatial map, for example, the order in which recruited neurons fire along a trajectory may produce an asymmetry in the interactions as a consequence of Spike Timing Dependent Plasticity (*Markram et al., 1997*), that requires the postsynaptic neuron to fire *after* the presynaptic one in order to strengthen the synapse. This phenomenon can be accounted for in the definition of the interaction kernel. Any asymmetric kernel can be decomposed in two contributions:

$$K(|\vec{x}_i - \vec{x}_j|) = K_S(|\vec{x}_i - \vec{x}_j|) + \gamma K_A(|\vec{x}_i - \vec{x}_j|) \tag{4}$$

where $K_S$ is the usual symmetric component and $K_A$ is an anti-symmetric function ($K_A(x_i - x_j) = -K_A(x_j - x_i)$). The parameter $\gamma$ regulates the relative strength of the two components. The presence of $K_A$ will generate a flow of activity along the direction of asymmetry: neuron $i$ activates neuron $j$ that, instead of reciprocating, will activate neurons downstream in the asymmetric direction. Mechanisms of this kind have been shown to produce a rigid shift of the encoded position along the manifold, without loss of coherence (*Zhang, 1996*; *Burak and Fiete, 2009*; *Fuhs and Touretzky, 2006*). In the quantitative analysis that follows we will concentrate, when not stated otherwise, on a kernel $K$ with the exponential form

$$K(|\vec{x}_i - \vec{x}_j|) = e^{-|\vec{x}_i - \vec{x}_j|} + \gamma \operatorname{sign}((\vec{x}_i - \vec{x}_j) \cdot \vec{n}) e^{-|\vec{x}_i - \vec{x}_j|/\xi} \tag{5}$$

where $\vec{n}$ is a unit vector pointing in the – constant – direction along which the asymmetry is enforced, and $\xi$ is the spatial scale of the asymmetric component, which is fixed to one where not explicitly stated otherwise. Moreover, we make the simplifying assumption of periodic boundary conditions on the manifold spanned by $\vec{x}$, such that the dynamics follows a periodic cycle. Both the kernel form and the periodic boundary conditions simplify the analytical description of the model, but are not a required feature of the model. In fact, all the results presented hold for a large class of interaction kernels, and the boundary conditions can be modified (e.g. with the introduction of interactions between different memories) without compromising the functionality of the network. Both points will be addressed in the analyses of the model in the next sections.

## Results

### Asymmetric recurrent connections produce dynamic retrieval

The spontaneous dynamics produced by the network is constrained to the low dimensional manifold codified in the connectivity matrix and spanned by the parameter $\vec{x}$. The short-range interactions and the uniform inhibition enforced by the firing threshold $h_0$ produce a localized 'bump' of activity in the manifold. The presence of an asymmetry in the connection strengths prevents the system from remaining in a stationary equilibrium. Instead, it generates a steady flow of activity in the direction of the asymmetry.

This flow is illustrated in *Figure 2 (a),(b) and (c)*, obtained with numerical simulation of a network encoding a one-, two-, or three-dimensional manifold, respectively. In the simulation, each neuron is assigned to a preferential firing location $\vec{x}_i$ in the manifold to encode, and the plots show the activity of the network organized according to this disposition. The activity of the population clusters in a

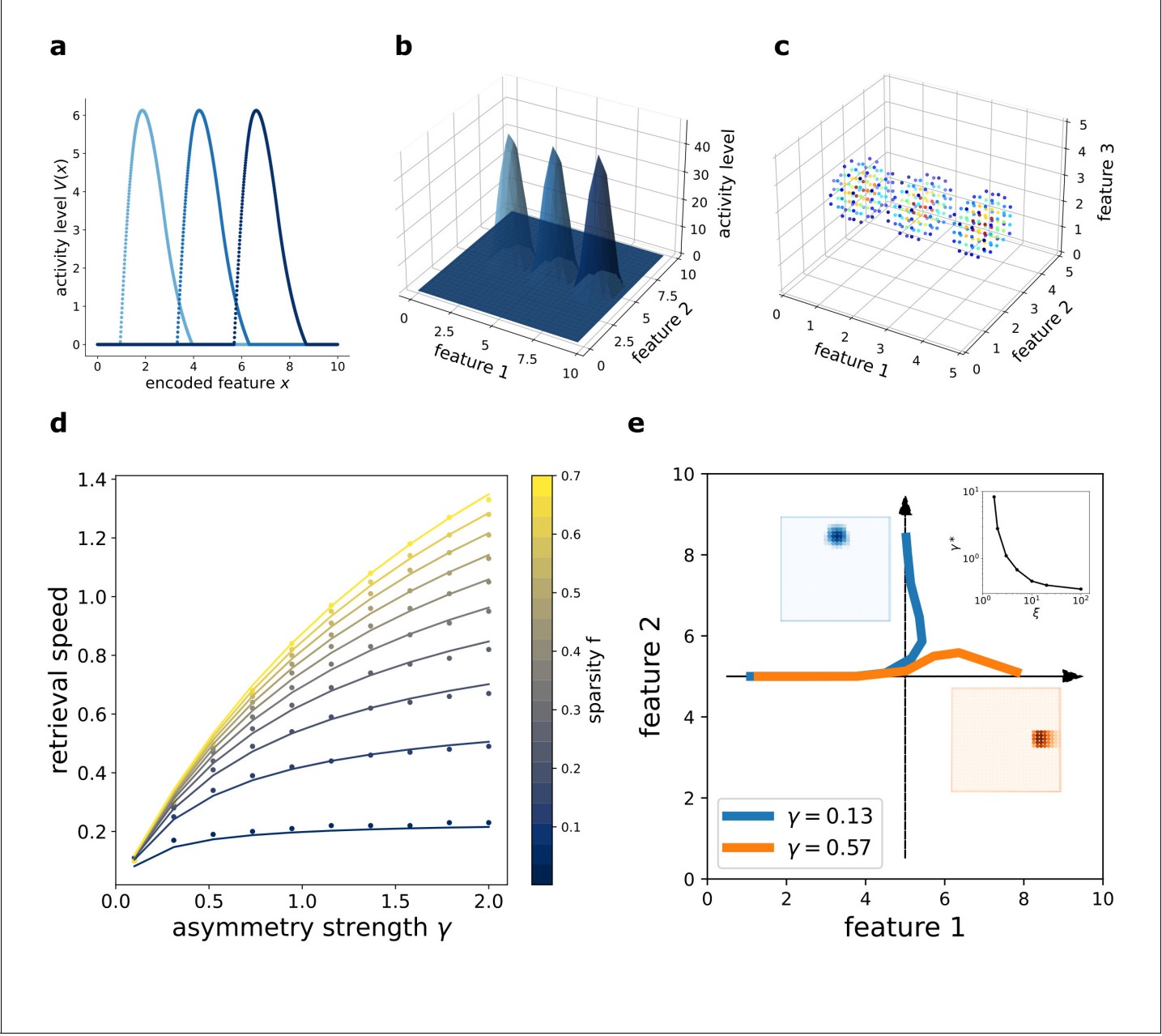

**Figure 2.** Dynamic retrieval of a continuous manifold. First row: each plot presents three snapshots of the network activity at three different times (t1, t2 and t3), for a system encoding a one dimensional (**a**), two dimensional (**b**) and three dimensional (**c**) manifold. In (**c**), activity is color-coded (blue represents low activity, red is high activity, silent neurons are not plotted for better readability). In all cases, the anti-symmetric component is oriented along the x axis. (**d**) Dependence of the speed on $\gamma$ and $f$. Dots are data from numerical simulations, full lines are the fitted curves. (**e**) Retrieval of two crossing trajectories. Black arrows represent the two intersecting encoded trajectories, each parallel to one of the axis. Full colored lines show the trajectories actually followed by the center of mass of the activity from the same starting point. Blue curve: low $\gamma$, the activity switches trajectories when it reaches the crossing point. Orange curve: high $\gamma$, successful crossing. In both cases $\xi = 10$. The blue and the orange insets show the activity bumps in the corresponding cases; the top-right inset shows the dependence of the value $\gamma^*$, required for crossing, on $\xi$.

bump around a certain position at each time point ($t_1$, $t_2$, and $t_3$), and the bump shifts by effect of the asymmetric component of the interactions. In this way, the neural population collectively encodes an evolving coordinate on the manifold spanned by $\vec{x}$. The coherence of the representation is not affected by the presence of the asymmetric term: the movement of the activity bump happens without dissipation.

The speed of movement of the bump is modulated by the value of the parameter $\gamma$ and by the sparsity level of the representation $f$, that is the fraction of neurons active at each given time during the dynamics. The dependence of the speed on these two parameters is illustrated in 2(d). Stronger asymmetry (high $\gamma$) produces a faster shift. Interestingly, the sparsity value $f$ acts as a modulator of the influence of $\gamma$: sparser representations move more slowly than dense ones.

While $\gamma$ describes a feature of the synaptic interactions, determined during the learning phase and relatively fixed at the short timescales of retrieval, $f$ can be instantaneously modulated during retrieval dynamics. A change in the gain or the excitability of the population can be used to produce dynamic retrieval at different speeds. Thus, the model predicts an interaction between the sparsity and the speed of the reactivation of a continuous memory sequence, with increased activity leading to faster replay. It is worth noting, however, that the *direction* of the dynamics is fixed with $J$: the model is able to retrieve either forward or backward sequences, but not alternate between them.

The dependence of the retrieval speed $s$ on $\gamma$ and $f$ is well described by the approximate functional form

$$s(\gamma, f) = \frac{A\gamma f}{b\gamma + cf + d\gamma f + e\gamma f^2} \tag{6}$$

This dependence is shown in 2(b), where the dots are the values obtained with numerical simulations and the full curves the fitted relationship. The full understanding of the nature of this functional form remains an open challenge for future analysis. As we will see in the next section, the analytical solution of the model yields the same form to describe the dependence of speed on $\gamma$ and $f$, but a closed-form solution is still lacking.

In the model presented, the asymmetry in the interactions is enforced uniformly along a single direction also for two- and three-dimensional manifolds, representing the case in which neural dynamics follows a forced trajectory along one dimension, but is free to move without energy costs along the others. However, the same mechanism can be used to produce one-dimensional trajectories embedded in low-dimensional manifolds, with the introduction of a positional dependence in the direction of the asymmetry (**Blum and Abbott, 1996**). In this case, an interesting problem is posed by the intersection of two trajectories embedded in the same manifold: is the network, during the retrieval of one trajectory, able to cross these intersections, or do they hinder dynamical retrieval? The investigation of the full phenomenology of position-dependent asymmetric kernels with intersecting trajectories is beyond the scope of the present work, but we present in *Figure 2(e)* a numerical study of a minimal version of this problem, with two orthogonal trajectories (*Figure 2(e)*, black arrows) embedded in a 2D manifold and memorized simultaneously in the network. Notice that in this case the two trajectories are parallel to the main axis of the square environment, but they do not need to be: any pair of orthogonal trajectories will behave in the same manner. When the network is cued to retrieve the horizontal trajectory, the behavior at the intersection depends on the strength $\gamma$ and scale $\xi$ of the asymmetric component. At low $\gamma$, the dynamics spontaneously switch trajectory at the intersection (*Figure 2(e)*, blue curve), while for $\gamma$ sufficiently large the retrieval of the horizontal trajectory is successful (*Figure 2(e)*, orange curve). The value $\gamma^*$ required for a successful crossing depends on the spatial scale $\xi$: larger $\xi$ allow for crossing with lower values of $\gamma$, as shown in the top-right inset of *Figure 2(e)*, in which $\gamma^*(\xi)$ is plotted. Intuitively, the ability of the network to retrieve crossing trajectories depends on the shape of the activity bump, which needs to be sufficiently elongated in the direction of retrieval for the successful crossing of the intersection. The blue and orange insets of *Figure 2(e)* show the difference in shape of the bump in the case of a trajectory switch (blue) and successful crossing (orange).

## Analytical solution for the single manifold case

The simplicity of continuous attractor models often allows to extract important computational principles from their analytical solution (**Wu et al., 2008**; **Fung et al., 2010**). In our case, the dynamic behavior of the system and its features can be fully described analytically with a generalization of the framework developed by **Battaglia and Treves, 1998**. For this purpose, it is easier to formulate the problem in the continuum, and describe the population activity $\{V_i\}$ by its profile $V(\vec{x})$ on the attractive manifold parametrized by the coordinate $\vec{x}$, and the dynamical evolution as a discrete step map, equivalent to *Equation 1*.

$$V(\vec{x}, t+1) = g[h(\vec{x}, t)]^+ \tag{7}$$

$$h(\vec{x}, t) = \int d\vec{x}' K(\vec{x} - \vec{x}') V(\vec{x}', t) - h_0 \tag{8}$$

The requirement of a rigid shift of population activity is then imposed by setting the activity at time $t + 1$ to be equal at the activity at time $t$, but translated by an amount $\Delta\vec{x} = \tau s \vec{n}$, proportional to the speed $s$ of the shift and in the direction $\vec{n}$ of the asymmetry in the connections. The timescale $\tau$ sets the time unit in which the duration of the evolution is measured and does not have an impact on the behavior of the system.

The activity profile $V(\vec{x})$ is then found as the self-consistent solution to the integral equation

$$V(\vec{x} + \Delta\vec{x}) = g\left[ \int d\vec{x}' K(\vec{x} - \vec{x}') V(\vec{x}') - h_0 \right]^+ \tag{9}$$

*Equation 9* is valid in general. We will focus here, for the explicit derivation (reported in appendix C), on the case of a one dimensional manifold with an exponential interaction kernel

$$K(x - x') = e^{-|x-x'|} + \gamma \operatorname{sign}(x - x') e^{-|x-x'|} \tag{10}$$

In this case, the activity bump will take the form:

$$V(x) = \begin{cases} Ce^{k_1 x} \cos(k_2 x) + \frac{g\theta}{1 - 2g} & \text{if } \text{-}R \le x \le R \\ 0 & \text{if } \text{-}R > x \text{ or } x > R \end{cases} \tag{11}$$

The parameters $k_1 = k_1(\gamma, s)$ and $k_2 = k_2(\gamma, s)$ determine the properties of the solution and they depend on the values of $\gamma$ and speed $s = \Delta x / \tau$.

$k_2$ is related to the bump width by the relation

$$R = \frac{\pi}{2k_2} \tag{12}$$

where $R$ is the point at which $V(x) = 0$. $k_1$ is related to the asymmetry of the bump: in the limit case $\gamma = 0$, $s = 0$ (*Figure 3(a)*, first column) $k_1 = 0$, and we recover the cosine solution of the symmetric kernel case studied in *Battaglia and Treves, 1998*. Larger $k_1$ values result in more and more asymmetric shapes (*Figure 3(a)*, second and third columns).

From this analytical solution, we can determine the dependence of the speed $s$ on the asymmetry strength $\gamma$ and on the sparsity $f = 2R/L$ (note that in the continuum case the fraction of active neurons is given by the ratio between the bump size $2R$ and the manifold size $L$). The sparsity $f$ is modulated by the value of the gain $g$, as shown in *Figure 3(b)*: a larger gain in the transfer function corresponds to a sparser activity. The exact relation $s(\gamma, f)$ can be obtained from the numerical solution of a transcendental equation (see appendix C), and can be approximated with a functional shape analogous to the one used for the simulated network:

$$s(\gamma, f) = \frac{A\gamma f}{b\gamma + cf + d\gamma f + e\gamma f^2} \tag{13}$$

The full transcendental solution and the fitted curves are reported in *Figure 3(c)*.

## Dynamic retrieval is robust

The analytical solution of the model shows that the network performs dynamical retrieval for all values of the asymmetry strength $\gamma$, and that this parameter influences the retrieval speed and the shape of the activity bump. To further investigate the robustness of dynamical retrieval to parameter changes, we investigate with numerical simulations the behavior of the model with respect to another important parameter: the scale $\xi$ of the anti-symmetric component. We run several dynamics of a network with interaction kernel given by

$$K(d) = e^{-d} + \gamma \operatorname{sign}(d) e^{-d/\xi} \tag{14}$$

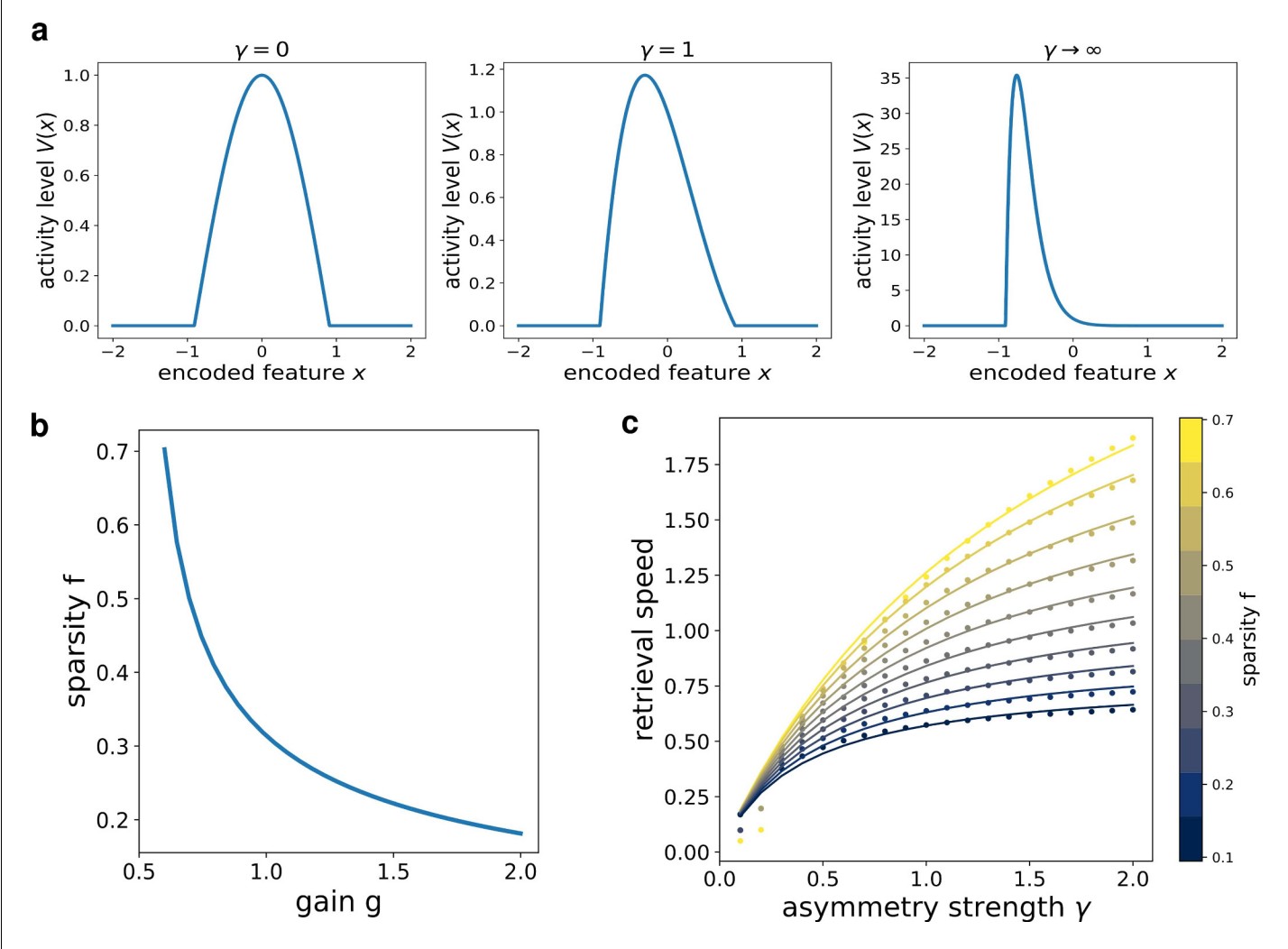

**Figure 3.** Analytical solution of the model. (**a**) The shape of the bump for increasing values of γ. (**b**) Dependence of the sparsity $f$ on the gain $g$ of the network. (**c**) Dependence of the speed of the shift on γ, at different values of sparsity. Dots show the numerical solution (note some numerical instability at low $f$ and γ), full curves are the best fits.

varying the parameters γ and ξ and measuring the retrieval speed, the peak value of the activity bump and its skeweness. The joint effects of γ and ξ are shown in *Figure 4*. In the whole range of parameters analyzed, spanning four orders of magnitude for both parameters, the network was able to produce dynamic retrieval. γ and ξ affect the speed of the shift, the peak values of the activity distribution and the skewness of the activity bump, without hindering network functionality. Moreover, all these feature vary gradually and mildly with the parameters values, producing dynamical behavior qualitatively similar in the full parameter range. This analysis shows that dynamical retrieval does not require any fine tuning of network parameters, but relies on the assumption of an exponential shape for the interaction kernel. How robust is the behavior of the network to the details of the kernel shape?

We addressed this question by simulating the network dynamics with alternative kernel choices. We kept fixed the symmetric component, and explored three different anti-simmetric shapes: a gaussian-derivative shape (*Figure 5a*), a sinusoidal shape (*Figure 5b*) and a double step function (*Figure 5c*). Each of these simulations produced the same retrieval dynamics (a stable bump shifting at constant speed), the only effect of the kernel shape being on the details of the shape of the activity bump (*Figure 5*, bottom row). This shows that the dynamic retrieval mechanism, much like standard continuous attractors, is robust with respect to the precise shape of the interactions.

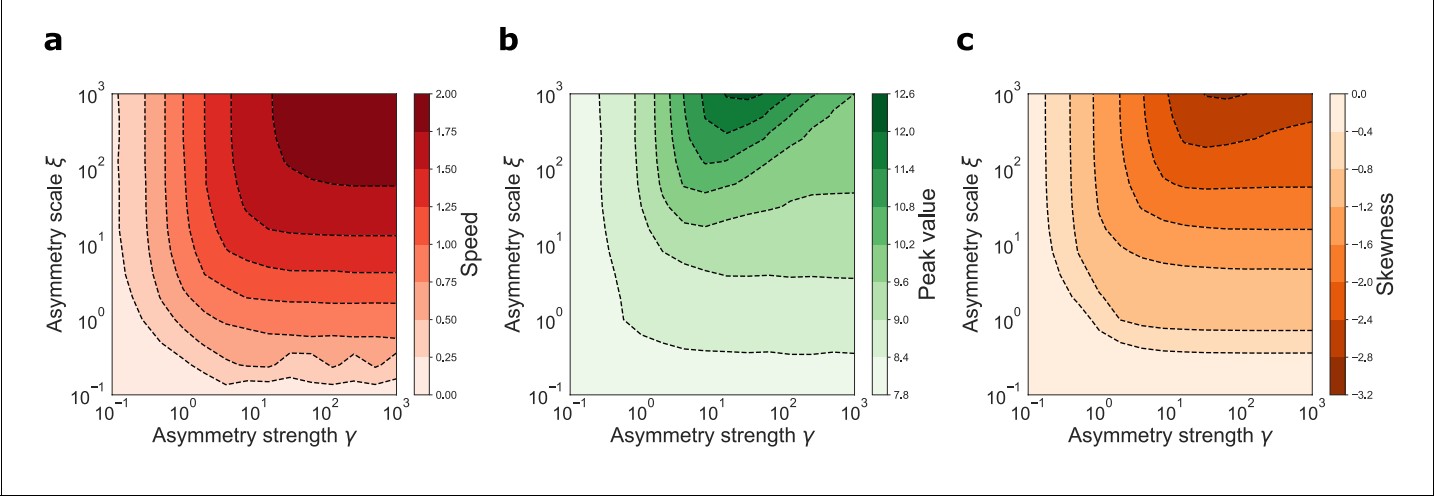

**Figure 4.** Dynamical retrieval in a wide range of parameters. Effect of the kernel strength γ and its spatial scale ξ, in the case of the exponential kernel $K(d) = e^{-|d|} + \gamma \mathrm{sign}(d) e^{-|d|/\xi}$ .(**a**) Retrieval speed (**b**) Peak value of the activity (**c**) Skewness of the activity bump.

Importantly, no particular relationship is required between the shape of the symmetric and the antisymmetric components of the kernel.

## A dynamical memory: storing multiple manifolds

We described in detail the behavior of a neural network with asymmetric connectivity in the case of a single manifold encoded in the synaptic connectivity. For the network to behave as an autoassociative memory, however, it needs to be able to store and dynamically retrieve *multiple* manifolds. This is possible if we construct the interaction matrix $J_{ij}$ as the sum of the contributions from $p$ different, independently encoded manifolds:

$$J_{ij} = \frac{1}{N} \sum_{\mu=1}^{p} K(\vec{x}_i^{\mu} - \vec{x}_j^{\mu}) \tag{15}$$

Here, each $x_i^{\mu}$ represent the preferred firing location of neuron $i$ in the manifold μ, and $K$ is the same interaction kernel as in *Equation 4*, containing a symmetric and anti-symmetric component.

The resulting dynamics show multiple continuous attractors, corresponding to the stored manifolds. Given an initial configuration, the networks rapidly converges to the nearest (i.e. most correlated) attractor, forming a coherent bump that then moves along the manifold as a consequence of the asymmetric component of the connectivity. The same dynamics, if projected on the other unretrieved manifolds, appear as random noise. This is illustrated in *Figure 6* obtained with numerical simulations of a network encoding three different manifolds (of dimension one in (a), dimension two in (b)), and dynamically retrieving the first one. How does this shifting activity bump relate to the activity of single cells? To clarify this aspect we simulated an electrophysiological recording from the dynamical retrieval of *Figure 6(a)*. We selected a random subset of 15 of the 1000 cells of the network, and generated spike trains using a poisson point process with an instantaneous firing rate proportional to the activity level of each cell yielded by the retrieval dynamics at each time step, plus a small noise. With this procedure we obtain the spike trains of each cell, that we can visualize with a rasterplot (*Figure 6c*). We can sort the cells according to their firing field position on each of the three manifolds (if these were, e.g., linear tracks, this would correspond to sorting according to place field positions in each of the tracks). When ordered as in the retrieved manifold, the recorded cells show a structured pattern that is considered the hallmark of sequential activity in experimental studies (*Figure 6c*, first column). If the same spikes are ordered according to the unretrieved manifolds, the pattern is lost (*Figure 6c*, second and third column), indicating that the population activity is retrieving the dynamical structure of the first manifold specifically.

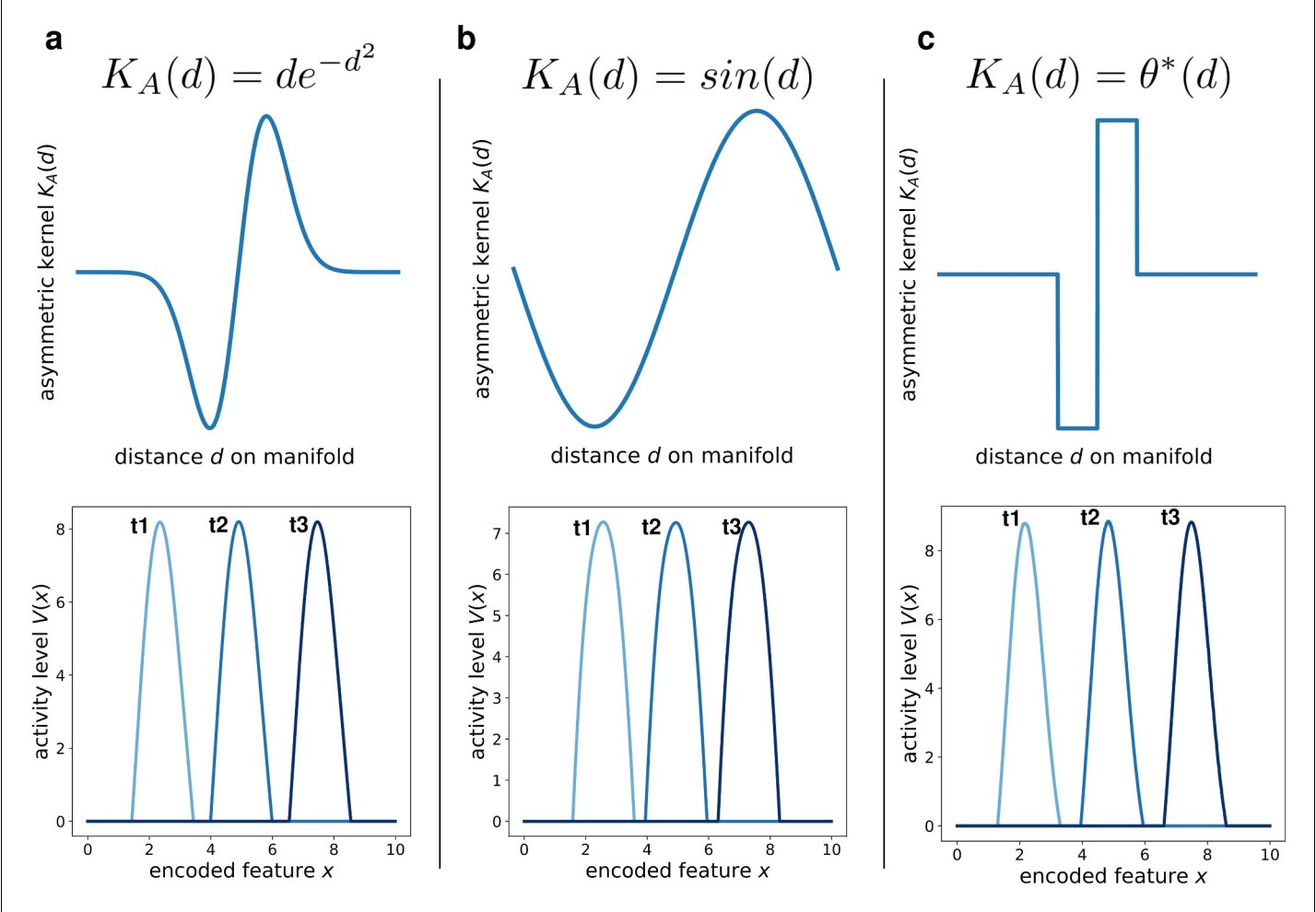

**Figure 5.** Different interaction kernels produce similar behavior. Three examples of dynamics with the same symmetric component and three different anti-symmetric components. Top row: shape of the anti-symmetric component $K_A$. Bottom row: three snapshots of the retrieval dynamics for the corresponding $K_A$. (a) Gaussian derivative; (b) Sinusoidal; (c) Anti-symmetric step function, $\theta^* = \theta(d)\theta(1-d) - \theta(-d)\theta(d-1)$.

Multiple dynamic manifolds can be memorized and retrieved by the network, with different speeds. *Figure 7 (b)* shows the result of the numerical simulation of a network with five different one-dimensional manifold stored in its connectivity matrix, each encoded with a different value of $\gamma$ (see appendix D). These manifold are dynamically retrieved by the network at different speeds, depending on the corresponding $\gamma$. This allows the model to simultaneously store memories without the constraint of a fixed dynamical timescale, an important feature for the description of biological circuits that need to be able to operate at different temporal scales.

Different memories stored in the same neural population can interact with each other, building more retrieval schemes in which, for example, the retrieval of a memory cues the retrieval of another one. To investigate this possibility, we have incorporated in the model a mechanism for interaction between memories, in which the endpoint of a dynamical, one-dimensional manifold elicits the activation of the start point of a different one (see Appendix E). This results in the sequential retrieval of multiple memories, one after the other, as illustrated in *Figure 7 (d)*. The top row shows the evolution in time of the overlaps $m_\mu$:

$$m_\mu(t) = \frac{1}{N^2} \sum_{i,j} K_S(x_i^\mu - x_j^\mu) V_i(t) V_j(t) \tag{16}$$

These order parameters quantify the coherence of the population activity $V(t)$ with each of the

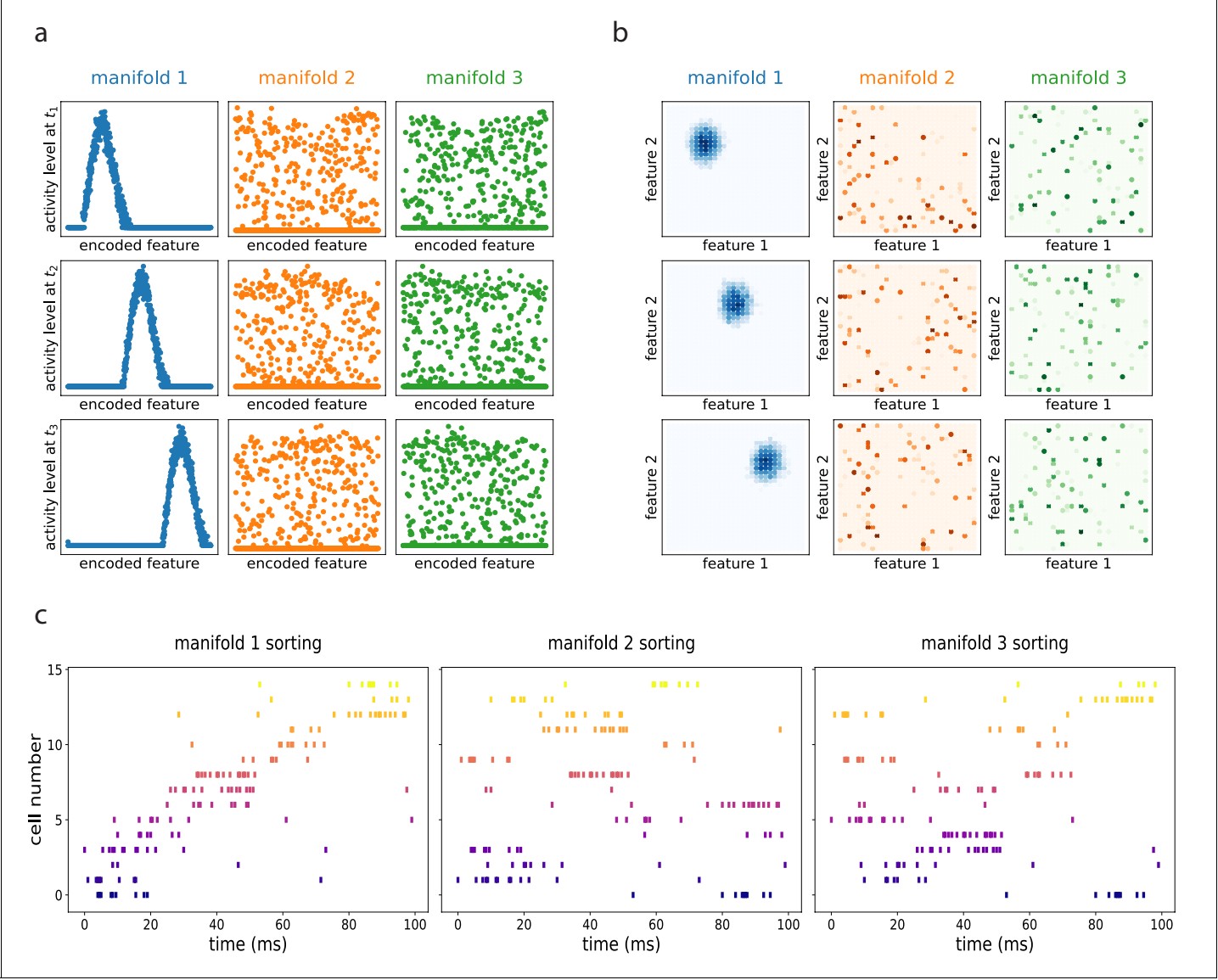

**Figure 6.** Dynamic retrieval in the presence of multiple memories. (**a**) In one dimension (**b**) In two dimensions. Each row represents a snapshot of the dynamics at a point in time. The activity is projected on each of the three attractors stored in the network. In both cases, the first attractor is retrieved, and the activity organizes in a coherent bump that shifts in time. The same activity, projected onto the two non-retrieved maps looks like incoherent noise ((**a**) and (**b**), second and third columns). (**c**) Spiking patterns from a simulated recording of a subset of 15 cells in the network. When cells are sorted according to their firing field on the retrieved manifold (first column), they show sequential activity. The same activity is scattered if looked from the point of view of the unretrieved manifolds (second and third columns).

manifolds. Localized activity in manifold μ results in a large $m_\mu$, while a low $m_\mu$ corresponds to an incoherent scattering of the activity. The network retrieves the manifolds in sequence, one at a time, following the instructed transitions encoded in its connectivity. The all-or-nothing behavior of the coherence parameters segments the continuous dynamics of the network into a sequence of discrete states.

The bottom row shows the evolution of the retrieved position, given in each manifold by the center of mass:

$$<x>_\mu (t) = \frac{1}{N} \sum_i x_i^\mu V_i(t) \tag{17}$$

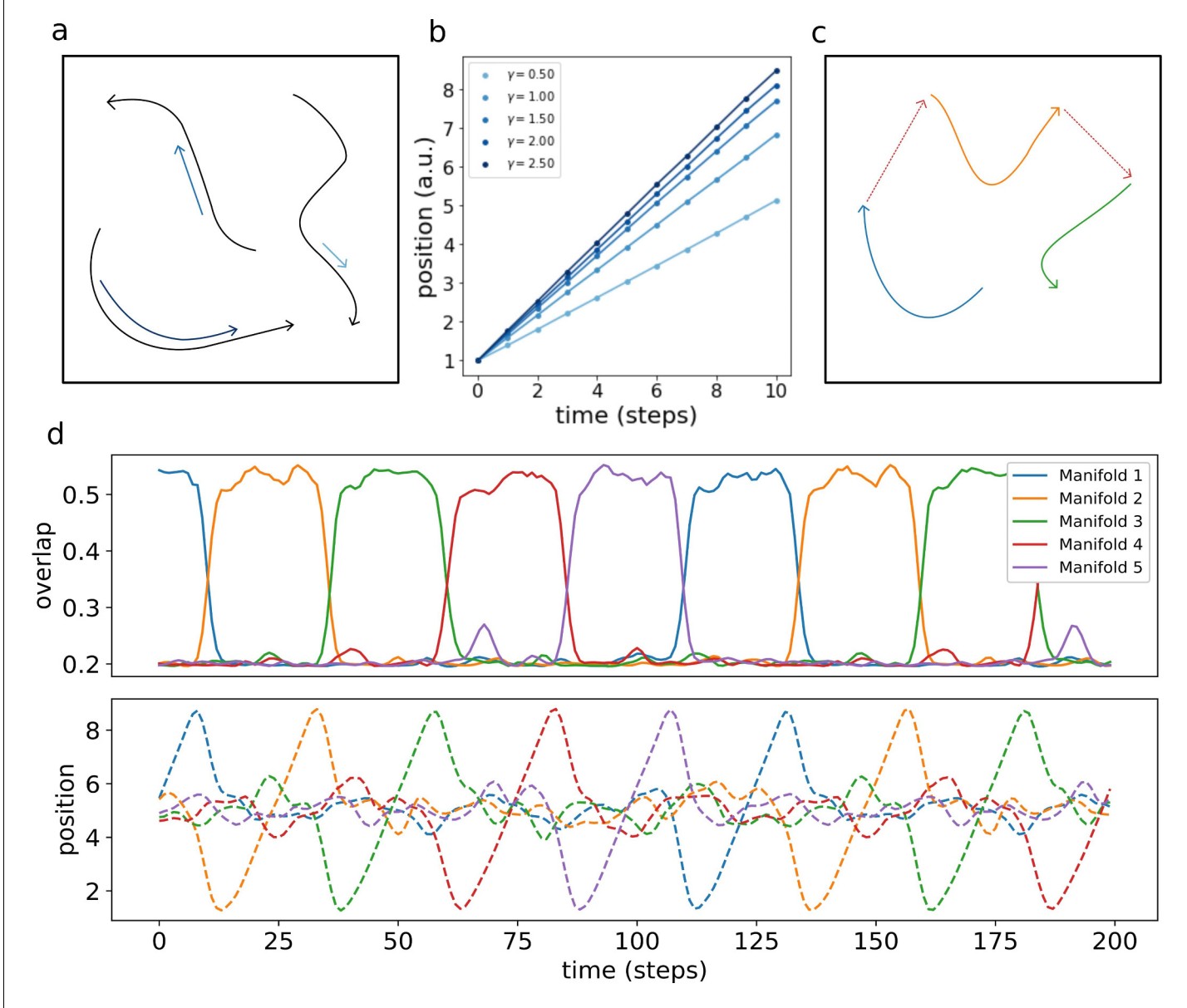

**Figure 7.** Retrieval speed and memory interactions. (a) Multiple mainfolds with different velocity can be stored in a network with manifold-dependent asymmetric connectivity (b) The retrieved position at different timesteps during the retrieval dynamics of five different manifolds, stored in the same network, each with a different value of γ. (c) Manifolds memorized in the same network can be linked together (d) Sequential retrieval of five manifolds. Top row: overlap, measuring the overall coherence with the manifold, as a function of time. Bottom row: retrieved position in each manifold as a function of time.

The dynamic runs across the retrieved manifold, from its beginning to its end, then jumps to next one and repeats the process. Note that the position in each of the un-retrieved manifold fluctuates around $L/2$, as a consequence of the incoherence of the activity. Within each of the retrieved manifolds, the dynamic retains its continuous nature in the representation of the evolving position.

This sequential dynamic goes beyond the simple cued retrieval of independent memories that is the focus of most autoassociative memory models, and provides an example of a hybrid computational system, encoding both continuous and discrete features.

The interaction mechanism introduced here provides the opportunity to investigate the effect of more complex interactions than the simple memory chain presented here. We present here this first example as a proof of principle of the possibility of storing interacting dynamical memories, and will

proceed to the investigation of more complex structures (e.g. interaction networks, probabilistic interactions, etc.) in future studies.

## Storage capacity

The number of maps that can be stored and retrieved by an attractor network of this kind is typically proportional to the number of inputs per neuron $C$ (*Treves and Rolls, 1991*). The memory load $\alpha = p/C$ crucially determines the behavior of the system: when $\alpha$ is increased above a certain threshold value $\alpha_c$, the network is not able to retrieve any of the stored memories, falling instead into a disordered state. Therefore it is the magnitude of $\alpha_c$, that is the storage capacity of the system, that determines how effectively it can operate as a memory. To estimate the storage capacity of dynamic continuous attractors, and to investigate how it is impacted by the presence of asymmetric connections, we proceed along two complementary paths.

In the case a fully connected network, where the analytical tools developed for equilibrium systems are not applicable, we take advantage of the fact that numerical simulations can be effective for the estimation of the capacity, since the number of connections per neuron $C$ (the relevant parameters in the definition of the storage capacity $\alpha_c = p/C$) coincides with the number of neurons, minus one. For a highly diluted system, on the other hand, the number of neurons is much larger than $C$, making the simulation of the system very difficult in practice. We then resort to an analytical formulation based on a signal-to-noise analysis (*Battaglia and Treves, 1998*), that exploits the vanishing correlations between inputs of different neurons in a highly diluted network, and does not require symmetry in the connectivity (*Derrida et al., 1987*). The quantification of the effect of loops in the dense connectivity regime, developed in *Shiino and Fukai, 1992* and *Roudi and Treves, 2004* for the case of static, discrete attractors, is beyond the scope of the present work and remains an interesting open direction.

In both the fully connected and the highly diluted case we study the dependence of the capacity on two important parameters: the map sparsity, that is the ratio between the width of the connectivity kernel (fixed to one without loss of generality) and the size $L$ of the stored manifolds, and the asymmetry strength $\gamma$. Note that the map sparsity $1/L$ is different from the activity sparsity $f$: the former is a feature of the stored memories, that we will treat as a control parameter in the following analysis; the latter is a feature of the network dynamics, and its value will be fixed by an optimization procedure in the calculation of the maximal capacity.

## Analytical calculation of the capacity in the highly diluted limit

The signal-to-noise approach we follow, illustrated in details in *Battaglia and Treves, 1998*, involves writing the local field $h_i$ as the sum of two contributions: a signal term, due to the retrieved – 'condensed' – map, and a noise term consisting of the sum of the contributions of all the other, 'uncondensed' maps. In the diluted regime ($C/N \to 0$), these contributions are independent and can be summarized by a Gaussian term $\rho z$, where $z$ is a random variable with zero mean and unit variance. In the continuous limit, assuming without loss of generality that map $\mu = 1$ is retrieved, we can write:

$$h(x^1) = g \int_L dx^{1\prime} K(x^1 - x^{1\prime}) V(x^{1\prime}) + \rho z \tag{18}$$

The noise will have variance:

$$\rho^2 = \alpha y L^2 \langle\langle K^2(x - x') \rangle\rangle \tag{19}$$

where $L$ is the size of the map, $\langle\langle K^2(x-x') \rangle\rangle$ is the spatial variance of the kernel and

$$y = \frac{1}{N} \sum_i V_i^2 \tag{20}$$

is the average square activity.

We can write the fixed point equation for the average activity profile $m^1(x)$, incorporating the dynamic shift with an argument similar to the one made for the single map case:

$$m^1(x + \Delta x) = g \int^+ Dz(h(x) - h_0) \tag{21}$$

where $Dz = (e^{-z^2/2}/\sqrt{2\pi})dz$ and $\int^+ f(x)dx = \int f(x)\theta(x)dx$. The average square activity $y$, entering the noise term, reads

$$y = \frac{g^2}{L} \int dx \int^+ Dz(h(x) - h_0)^2 \tag{22}$$

Introducing the rescaled variables

$$w = \frac{-h_0}{\rho} \tag{23}$$

$$v(x) = \frac{m^1(x)}{\rho} \tag{24}$$

And the functions

$$\mathcal{N}(x) = x\Phi(x) + \sigma(x) \tag{25}$$

$$\mathcal{M}(x) = (1 + x^2)\Phi(x) + x\sigma(x) \tag{26}$$

where $\Phi(x)$ and $\sigma(x)$ are the Gaussian cumulative and the Gaussian probability mass function respectively, we can rewrite the fixed-point equation as

$$v(x + \Delta x) = g\mathcal{N}\left(\int dx' K(x - x')v(x') + w\right) \tag{27}$$

$$y = \rho^2 g^2 \int \frac{dx}{L} \mathcal{M}\left(\int dx' K(x - x')v(x') + w\right) \tag{28}$$

Substituting *Equation 28* in the expression for the noise variance 19 we obtain

$$\frac{1}{\alpha} = g^2 L\langle\langle K^2\rangle\rangle \int dx \mathcal{M}\left(\int dx' K(x - x')v(x') + w\right) \tag{29}$$

If we are able to solve *Equation 27* for the rescaled activity profile $v(x)$, we can use *Equation 29* to calculate $\alpha$. We can then maximize $\alpha$ with respect to $g$ and $w$: this yields the maximal value $\alpha_c$ for which retrieval solutions can be found.

These equations are valid in general and have to be solved numerically. Here we present the results for the case of one-dimensional manifolds and interactions given by the exponential kernel of *Equation 36*. In this case, we have

$$\langle\langle K^2(x - x')\rangle\rangle = (1 + \gamma^2)\langle\langle K_S^2(x - x')\rangle\rangle. \tag{30}$$

where $K_S(x - x') = e^{-|x-x'|}$ is the symmetric component of the kernel. A simple approximation, illustrated in appendix F along with the detailed solution procedure, allows to decouple the dependence of $\alpha_c$ on $\gamma$ and $L$, with the former given by the spatial variance given by *Equation 30* and the latter by the solution of *Equations 27 and 29* in the $\gamma = 0$ case. We therefore have:

$$\alpha_c(L, \gamma) \sim \alpha_c(L, 0)/(1 + \gamma^2) \tag{31}$$

The storage capacity is plotted in *Figure 8(a)* as a function of $\gamma$ and $L$.

For sparse maps and small values of the asymmetry, the capacity scales as

$$\alpha_c \sim -\frac{1}{\ln(1/L)(1 + \gamma^2)} \tag{32}$$

The scaling with $1/L$ is the same found by *Battaglia and Treves, 1998* in the analysis of the symmetric case, as expected: for $\gamma = 0$ the two models are equivalent.

The presence of asymmetry decreases the capacity, but does not have a catastrophic effect: the decrease is continuous and scales with a power of $\gamma$. There is therefore a wide range of values of

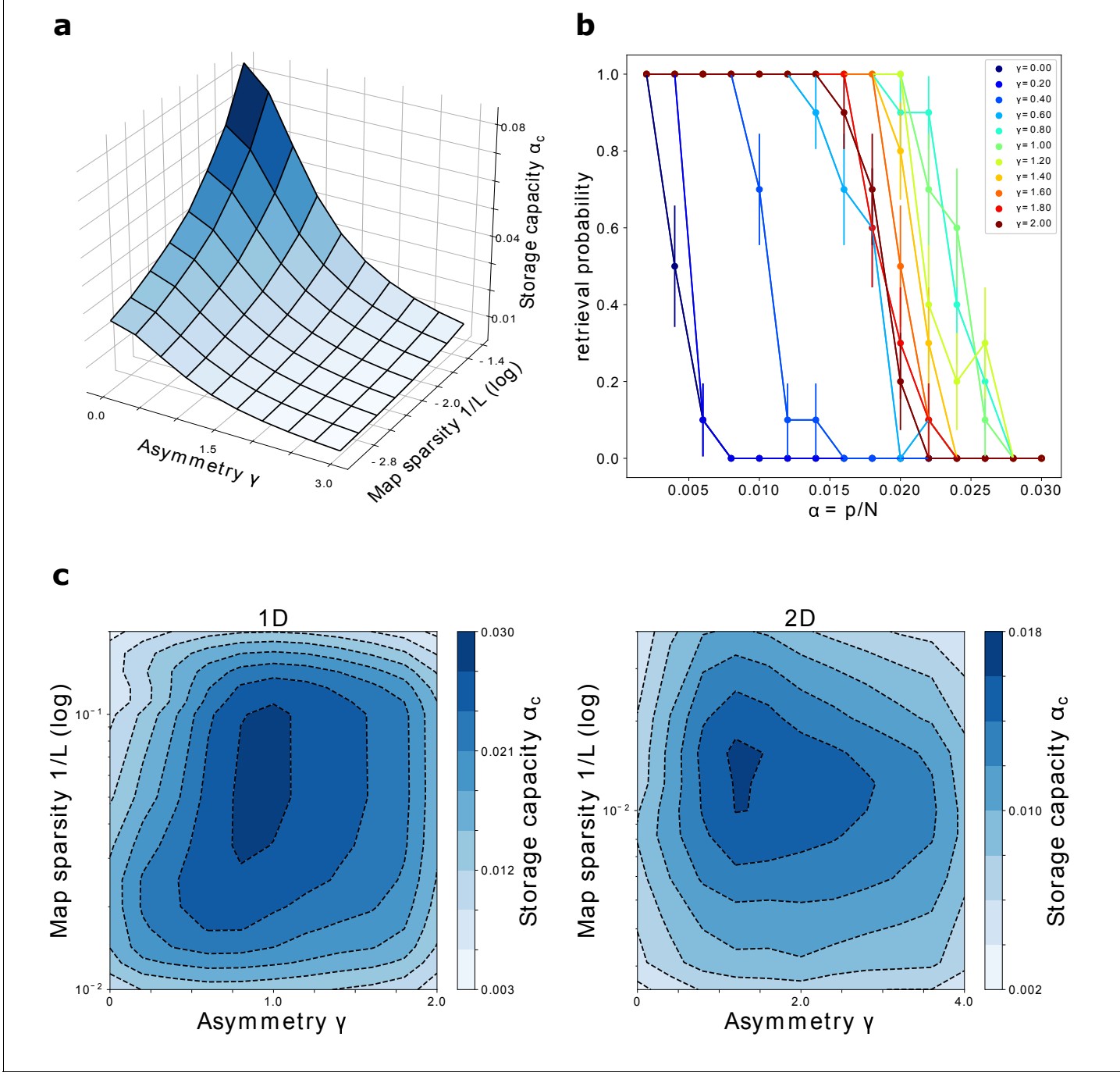

**Figure 8.** Storage capacity. (a) Storage capacity of a diluted network: dependence on $\gamma$ and $1/L$ (represented as $log_{10}(1/L)$). (b) Storage capacity of a fully connected network: non monotonic dependence of the capacity on $\gamma$. Retrieval / no retrieval phase transition for different values of $\gamma$, obtained from simulations with $N = 1000$, $N_S = 10$ and $L = 10$, for 1D manifolds. Error bars show the standard error of the observed proportion of successful retrievals. The non-monotonic dependence of the capacity from $\gamma$ can be appreciated here: the transition point moves toward the right with increasing $\gamma$ up to $\gamma \sim 1$, then back to the left. (c) Storage capacity of a fully connected network as a function of map sparsity $1/L$ and asymmetry strength $\gamma$, for a one-dimensional and a two-dimensional dynamic continuous attractor.

asymmetry and map sparsity in which a large number of dynamic manifolds can be stored and retrieved.

## Numerical estimation of the capacity for a fully connected network

To estimate the storage capacity for a fully connected network, we proceed with numerical simulations. For a network of fixed size $N$, and for given $\gamma$, $L$ and number of maps $p$, we run a number of simulations $N_S$, letting the network evolve from a random initial configuration. We consider a simulation to have performed a successful retrieval if the global overlap

$$m^\mu = \frac{1}{N^2} \sum_{i \neq j} V_i V_j K_S(x_i^\mu - x_j^\mu) \tag{33}$$

that quantifies the coherence of the activity with map $\mu$, is large for one map $\mu^*$ (at least 95% of the overlap value obtained in the case of a single map) and low in all others maps $\mu \neq \mu*$. We then define the retrieval probability as $p_r = N_R/N_S$, where $N_R$ is the number of observed retrievals.

We repeat the process varying the storage load, that is the number of stored manifolds $p$. As $p$ is increased, the system reaches a transition point, at which the retrieval probability rapidly goes to zero. This transition is illustrated, for various values of $\gamma$, in *Figure 8(b)*.

The number of maps $p_c$ at which the probability reaches zero defines the storage capacity $\alpha_c(\gamma, L) = p_c(\gamma, L)/N$. Repeating this procedure for a range of values of $\gamma$ and $L$, we obtain the plots shown in *Figure 8(c)*, for networks encoding one dimensional and two dimensional dynamical memories.

The first thing that can be noticed is that the network can store a large number of maps in the fully connected case as well, for a wide range of $\gamma$ and $L$. A network with size in the order of ten thousand neurons could store from tens up to hundreds of dynamical memories.

The capacity for one dimensional attractors is higher than the one for their two dimensional counterparts. This is in line with what was found for symmetric networks (*Battaglia and Treves, 1998*).

Finally, we see that the peak of the capacity is found not only for intermediate values of map sparsity – again in line with what is known from the symmetric case – but also for intermediate values of the coefficient $\gamma$. This shows that moderate values of asymmetry can be beneficial for the storage of multiple continuous attractors, a non-trivial phenomenon that may be crucial for the memory capacity of biological networks. In particular this suggests that the natural tendency of the neural activity to show a rich spontaneous dynamics not only does not hinder the possibility for multiple memories to coexist in the same population, but can be a crucial ingredient for the correct functioning of memory mechanisms.

## Discussion

The results presented show how a continuous attractor neural network with memory-dependent asymmetric components in the connectivity can function as a dynamic memory. Our model is simple enough to be treated analytically, robustly produces dynamic retrieval for a large range of the relevant parameters and shows a storage capacity that is comparable to – and in some cases higher than – the capacity for static continuous attractors.

The analytical solution of the single manifold case shows that the interaction between the strength of the asymmetry and the velocity of the shift can be modulated by global features of the network activity such as its sparsity. This makes the network able to retrieve at different velocities in different regimes, without necessarily requiring short term synaptic modifications. The dependence of the retrieval speed from the sparsity of the activity yields a testable prediction in the context of hippocampal replay: faster reactivations are to be expected in association with an increase in the excitability of the population.

The insensitivity of the general features of the dynamics to the fine details of the shape of the interactions suggests that this mechanism could robustly emerge from learning or self organization processes in the presence of noise. The quantitative analysis of the learning process needed to effectively memorize low-dimensional dynamic manifolds is an interesting open direction, which goes beyond the scope of this work. However, the asymmetric Hebbian plasticity rule used here provides a simple and biologically realistic starting point.

Our analysis shows that the simple introduction of an aysmmetric Hebbian plasticity rule is sufficient to describe a dynamic memory able to store and retrieve multiple manifolds with different

speeds, and it can incorporate interactions between them, producing the chained retrieval of a sequence of continuous memories.

The central result of the paper is the quantification of the storage capacity for dynamic continuous attractors, that we find to be large in magnitude, only mildly impacted by the asymmetry in diluted networks, and even higher than the capacity for static attractors in fully connected networks with moderate degrees of asymmetry. The storage capacity of out of equilibrium continuous attractors has been calculated, in a different scenario, by *Zhong et al., 2020*. The authors considered the case of an external signal driving the activity bump along the attractor, in a network of binary neurons, and proceeded to calculate the storage capacity with several assumptions that allowed to model the interference of multiple maps as thermal noise. Interestingly, their main result is broadly compatible with what we show here: in the highly diluted regime the velocity of the external signal has a mild – detrimental – effect on the capacity. This hints that out of equilibrium effects could show some form of universality across different network models and implementations of the shift mechanism. Moreover, a high capacity for dynamical sequences has shown to be achievable also in the case of discrete items (*Gillett et al., 2020*). Together these results suggest that the introduction of a temporal structure is compatible with the functioning of autoassociative memory in recurrent networks, and they open the way to the use of attractor models for the quantitative analysis of complex memory phenomena, such as hippocampal replay and memory schemata.

The model we propose suggests that the tendency of the activity to move in the neural population is a natural feature of networks with asymmetric connectivity, when the asymmetry is organized along a direction in a low dimensional manifold, and that static memories could be the exception rather than the rule. Indeed, *Mehta et al., 2000* have shown that place fields can become *more* asymmetric in the course of spatial learning, demonstrating that the idea that symmetry emerges from an averaging of trajectory-dependent effects (*Sharp, 1991*) does not always hold true. The structural role of the asymmetry has important implications for the analysis methods used to describe the activity of large populations of neurons, which often rely on the assumption of symmetry in the interactions (e.g. in the analysis of pairwise correlations) or equilibrium of the neural activity (e.g. the standard inverse Ising inference).

In most of the two- and three-dimensional cases analysed here, the asymmetry is constant along a single direction in each attractor. This can describe the situation in which the temporal evolution of the memory is structured along a certain dimension, and free to diffuse, without energy costs, in the remaining ones. The description of several one-dimensional trajectories embedded in a two dimensional or three dimensional space requires a position-dependent asymmetric component. A systematic analysis of this situation is left for future analysis. However, the simple case of two intersecting trajectories embedded in a 2D map, analysed here, provides a proof of concept that several intersecting trajectories can be correctly retrieved, provided that the activity bump is sufficiently elongated in the direction of the trajectory. A progressive elongation of the place fields in the running direction has been observed in rats running on a linear track (*Mehta et al., 1997*), and our analysis predicts that an analogous effect would be observed also in open-field environments, when restricting the analysis to trajectories in the same running direction.

Another challenge is posed by the evidence that replayed sequences can be organized both forward and backward in time (*Foster and Wilson, 2006*). The model in its current formulation can produce the retrieval of a given sequence either forward *or* backward, but cannot alternate between the two. This suggests that, if replay relies on asymmetric connections, the hippocampus would have to use different representations for the forward and the backward component. The fact that a change in reward uniquely modulates backward replay (*Ambrose et al., 2016*) provides some evidence in this direction, but this question remains open to experimental investigation.

The dynamical retrieval of the model generalizes, in the framework of attractor networks, the idea of cognitive maps, incorporating a temporal organization in the low-dimensional manifold encoding the structure of the memory. This feature is reminiscent of the idea of memory schemata – constructs that can guide and constrain our mental activity when we reminisce about the past, imagine future or fictional scenarios or let our minds free to wander (*Ciaramelli and Treves, 2019*). The use of the present model to describe memory schemata will require further steps, such as an account of the interaction between hippocampus and neocortex, and the expansion of the mechanism describing the transition between different dynamical memories. Nevertheless, the idea of dynamic retrieval of

a continuous manifold and the integration of the model presented here with effective models of cortical memory networks (*Boboeva et al., 2018*) open promising perspectives.

Finally, the full analytical description of a densely connected asymmetric attractor network is a challenge that remains open, and can yield valuable insights on the workings of the neural circuits underlying memory.

## Acknowledgements

Work supported by the Human Frontier Science Program RGP0057/2016 collaboration. We are grateful for inspiring exchanges with Remi Monasson and others in the collaboration, and thank Silvia Girardi for her help with *Figure 1*.

## Additional information

### Funding

| Funder | Grant reference number | Author |
| --- | --- | --- |
| Human Frontier Science Program | RGP0057/2016 | Alessandro Treves |

The funders had no role in study design, data collection and interpretation, or the decision to submit the work for publication.

### Author contributions

Davide Spalla, Conceptualization, Software, Formal analysis, Investigation, Visualization, Methodology, Writing - original draft; Isabel Maria Cornacchia, Software, Validation, Investigation, Visualization, Methodology; Alessandro Treves, Conceptualization, Supervision, Funding acquisition, Validation, Investigation, Writing - review and editing

### Author ORCIDs

Davide Spalla  https://orcid.org/0000-0002-0328-6476
Isabel Maria Cornacchia  http://orcid.org/0000-0002-0704-7480
Alessandro Treves  http://orcid.org/0000-0001-7246-5673

### Decision letter and Author response

Decision letter https://doi.org/10.7554/eLife.69499.sa1
Author response https://doi.org/10.7554/eLife.69499.sa2

## Additional files

### Supplementary files

- Transparent reporting form

### Data availability

The work did not generate any experimental dataset. The code used for numerical simulations is publicly available on Github (https://github.com/davidespalla/CADM (copy archived at https://archive.softwareheritage.org/swh:1:rev:65a5bdfa291840cf5bf10e1da48aadb0b316a445)).

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

## Appendix 1

### A Numerical simulations

Numerical simulations are performed with python code, available at: https://github.com/davides-palla/CADM (copy archived at swh:1:rev:65a5bdfa291840cf5bf10e1da48aadb0b316a445), *Spalla, 2021*.

In the single map case, to each of the $N$ units ($N = 1000$ in 1D, $N = 1600$ in 2D) is assigned a preferential firing location $x_i$ on a regular grid spanning the environment with linear dimension $L$. From this preferred firing locations the interaction matrix $J_{ij}$ is constructed, with the formula:

$$J_{ij} = K_S(|x_i - x_j|) + \gamma K_A(|x_i - x_j|) \tag{34}$$

The precise shape of the symmetric and anti-symmetric parts of the kernel are chosen differently in different simulations, according to the feature the analysis focused on, as specified in the main text. Once the network is assembled, the dynamics is initialized either with a random assignment of activity values to each unit in the range $[0,1]$, or with a gaussian bump centered in the middle of the environment (note that, due to the periodic boundary conditions and the translational invariance of the connectivity, the choice of the starting point does not influence the outcome). The dynamics is then evolved in discrete time steps, with the iteration of the following operations:

- Calculation of the local fields $h_i(t) = \sum_j J_{ij} V_j(t-1)$
- Calculation of the activity values $V_i(t) = g(h_i(t) - h_0)\theta(h_i(t) - h_0)$
- Dynamic adjustment of the threshold $h_0$ such that only the $fN$ most active neurons remain active: $h_0 = V_f$, where $\sum_{j,V_j > V_f} = Nf$
- Recalculation of the activity $V_i(t)$ with the adjusted threshold
- Dynamic adjustment of the gain $g$ such that the mean activity $\langle\{V_i(t)\}\rangle$ is fixed to 1: $g = g/\langle\{V_i(t)\}\rangle$
- Recalculation of the activity $V_i(t)$ with the adjusted gain

The adjustment of the parameters of the transfer function is enforced to constrain the network to operate at fixed sparsity $f$ and fixed mean, set to one without loss of generality. The dynamics is iterated for a given number of steps (usually 200), large enough to assure the convergence to the attractive manifold (reached usually in < five steps) and the observation of the dynamical evolution on the manifold.

In the case of multiple maps, the implemented dynamical evolution is the same, but the interaction matrix is constructed with multiple assignments of the preferred firing locations $x_i$, one for each of the $p$ stored maps:

$$J_{ij} = \sum_{\mu=1}^{p} \left( K_S(|x_i^\mu - x_j^\mu|) + \gamma K_A(|x_i^\mu - x_j^\mu|) \right) \tag{35}$$

The multiple assignments of the preferred firing locations are performed by a random shuffling of the labels of the units before the assignment to the position on the regular grid spanning each map.

### B Simulation of electrophysiological recordings

To simulate the recording from a subset of the network during dynamical retrieval, we constructed a network of 1000 neurons with three different manifolds encoded in its connectivity matrix. We simulated a dynamic of the network from an initial condition correlated with the first manifold, for $T = 400$ time steps. Since we used circular manifolds, we then selected a chunk of the dynamics corresponding to a single lap on the circle, to have an easier scenario to compare with experimental work. We then simulated an experimental recording by selecting a random subset of 15 observed cells. We used the activity values of each cell during the dynamics as the instantaneous firing rate of a Poisson random process to generate the spiking activity, using a conversion of $1/25$ to convert the arbitrary value of activity to a plausible range of firing rates, in the order of some Hertz. Spikes produced by this process are our simulated activity recording, and can be plotted according to the preferred firing position of each of the recorded cell in each of the three manifolds. The preferred firing position is supposed to be extracted, in an experimental setting, from the average rate maps of the

recorded cells over many observations of the dynamics. In this way we obtain the plots reported in *Figure 6(c)*.

## C Analytical solution of the single map model in one dimension

To solve the integral *Equation 9* in the case of a 1D manifold and the exponential kernel

$$K(x-x') = e^{-|x-x'|} + \gamma \text{sign}(x-x') e^{-|x-x'|} \tag{36}$$

we start by rewriting it as

$$V(x+\Delta x) = \begin{cases} g \int_{-R}^{R} dx' K(x-x') V(x') - h_0, & \text{if } x \in \Omega \\ 0 & \text{otherwise} \end{cases} \tag{37}$$

where $[-R, R], R{>}0$ is a compact domain for which there exist a solution of *Equation 9* that is zero at the boundary. This allows to exploit the fact that our threshold-linear system is, indeed, linear in the region in which $V(x){>}0$.

We then differentiate twice to obtain the differential equation

$$V''(x+\Delta x) = V(x+\Delta x) + 2gV(x) + 2g\gamma V'(x) + g\theta \tag{38}$$

This is a second order linear ODE, with constant coefficients. The presence of the shift term $\Delta x$ inside the unknown function makes the equation non-trivial to solve. To solve it, we proceed in the following way: first, we look for a particular solution, that is easily found in the constant function

$$V_c = \frac{g\theta}{1-2g} \tag{39}$$

Then, we consider the associated homogeneous equation, and look for a solution in the form $V(x) = e^{kx}$, where $k$ is a solution of the characteristic equation $C(k) = 0$, with

$$C(k) = k^2 e^{k\Delta x} + 2g\gamma k + 2g - e^{k\Delta x}. \tag{40}$$

This trascendental equation has to be solved graphically in the complex domain, as shown in *Appendix 1—figure 1*.

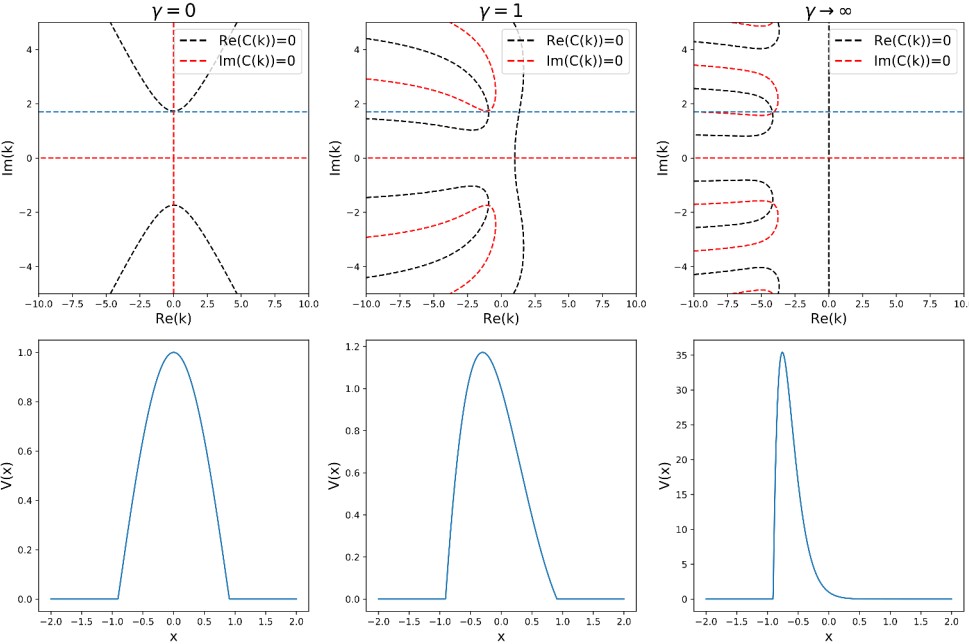

*Appendix 1—figure 1 continued on next page*

**Appendix 1—figure 1.** Analytic solution of *Equation (38)*. The top row shows the graphical procedure to find the complex zeros of the characteristic $C(k)$ given in (40), for three different values of $\gamma$. Black and red lines show the zeros of the real and imaginary part of $C(k)$, respectively. Their intersections are the complex solutions to $C(k) = 0$. The blue line represents the sparsity constraint $k_i = \pi/2R$. The bottom row shows the corresponding solution shapes.

For each value of $\gamma$ and $\Delta x$, the equation shows a pair of complex conjugate solutions

$$k_{1,2}^*(\gamma, \Delta x) = k_r(\gamma, \Delta x) \pm i k_i(\gamma, \Delta x) \tag{41}$$

The general solution of the equation will therefore have the form

$$V(x) = \begin{cases} Ce^{k_r x}\cos(k_i x) + \frac{g\theta}{1-2g} & \text{if } -R \leq x \leq R \\ 0 & \text{if } -R > x \text{ or } x > R \end{cases} \tag{42}$$

From *Equation 42* we can see that the absolute value of $k_i$ is related to the width of the bump, and therefore to the sparsity of the solution, by the relation

$$R = \frac{\pi}{2k_i}. \tag{43}$$

$R$ does in turn depend only on the free parameter $g$, through the relation that can be derived in the symmetric case ($\gamma = 0$, $\Delta x = 0$)

$$\tan(\sqrt{2g-1}R) = -\sqrt{2g-1} \tag{44}$$

We can look for a solution with given sparsity $f = 2R/L$ (where $L$ is the length of the manifold) by setting $R = fL/2$. Requiring the the sparsity to be fixed (i.e. setting $R$ constant) while varying $\gamma$ constrains the zeros of 40 to lie in the subspace $k_i = \pi/2R$. This imposes a relation between $\gamma$ and both the speed $s = \Delta x/\tau$ (related to the speed of the shift) and $k_r$ (related to the asymmetry of the shape of the solution). Varying $R$ we can study the dependence of the speed on both $\gamma$ and $f$.

## D Storing multiple manifolds with different retrieval speeds

To investigate the possibility to store and dynamically retrieve manifolds at different speeds, we have performed numerical simulations of a network with recurrent connectivity given by the formula

$$J_{ij} = \frac{1}{N}\sum_{\mu=1}^{p} \frac{1}{1+\gamma_\mu}\left[K_S(x_i^\mu - x_j^\mu) + \gamma_\mu K_A(x_i^\mu - x_j^\mu)\right] \tag{45}$$

Each manifold $\mu$ is encoded with a different asymmetry strength $\gamma_\mu$. The normalization factor $1/(1+\gamma_\mu)$ is added to ensure that each manifold contributes equally to the synaptic efficacies, and does not affect the ratio of strenghts of the symmetric and asymmetric components.

## E Linking multiple manifolds together

In order to model the interaction between different stored manifolds, we add to the connectivity matrix an "heteroassociative' term, whose strength $J_{H_{ij}}^{\mu\nu}$ is proportional to the distance between the preferred firing location $x_i^\mu$ of neuron $i$ in manifold $\mu$ and the one of neuron $j$ in manifold $\nu$, shifted by the length $L$ of the first manifold, which we denote by $\tilde{x}^\nu$. This shift enforces the fact that the interaction happens between the end of the first manifold and the beginning of the second. Then, the connectivity matrix will be given by

$$J_{ij} = \frac{1}{N}\sum_{\mu=1}^{p}K(x_i^\mu - x_j^\mu) + \sum_{\mu\nu}G_{\mu\nu}K(x_i^\mu - \tilde{x}_j^\nu) \tag{46}$$

With

$$\tilde{x}^{\nu} = x^{\nu} + L \tag{47}$$

and $G_{\mu\nu} = 1$ if there is a transition between $\mu$ and $\nu$, and zero otherwise.

## F Analytical calculation of $\alpha_c$ in the highly diluted limit

To calculate the maximum capacity, we first need to solve *Equation 27* numerically for $v(x)$, for given $g$ and $w$. The procedure illustrated here focuses, for the sake of analytical simplicity, on the case of a one-dimensional, exponential kernel

$$K(x - x') = e^{-|x-x'|} - \gamma \text{sign}(x - x') e^{-|x-x'|} \tag{48}$$

We start from *equation 27*:

$$v(x + \Delta x) = g\mathcal{N}\left(\int dx' K(x - x') v(x') + w\right) \tag{49}$$

First, following *Battaglia and Treves, 1998* we rewrite it with the transformation

$$u(x) = \mathcal{N}^{-1}\left(\frac{v(x)}{g}\right) \tag{50}$$

obtaining

$$u(x + \Delta x) = g\int dx' K(x - x')\mathcal{N}(u(x')) + w. \tag{51}$$

We then transform this integral equation in a differential one, by differentiating twice. We obtain

$$u''(x + \Delta x) + 2g\gamma\Phi(u(x))u'(x) + 2g\mathcal{N}(u(x)) - u(x + \Delta x) + w = 0 \tag{52}$$

where we have used the fact that $\mathcal{N}'(x) = \Phi(x)$. *Equation 52* is a second order, nonlinear delayed differential equation. To solve it, it is not sufficient to impose an initial condition on a single point for the solution and the first derivative (i.e. something like $u(x_0) = u_0, u'(x_0) = u'_0$): we have to specify the value of the function and its derivative in an interval $[x_0, x_0 + \Delta x]$.

To do so, we reason that, if we want a bump solution, $u(x)$ has to be finite for $x \to \pm\infty$ and cannot diverge. We then require the function to be constant ($u(x) = u_0$, $u'(x) = 0$) before a certain value $x_0$, whose value can be set arbitrarily without loss of generality.

The value $u_0$, at $\gamma = 0$ and $\Delta x = 0$ determines the shape of $u(x)$, as shown by the numerical solution presented in *Appendix 1—figure 2*. For $u_0 < u^*$ the solution will diverge at $x \to \infty$, while for $u_0 > u^*$ it will oscillate. We are then left with a single value $u_0(g, w) = u^*(g, w)$ for which the solution has the required form.

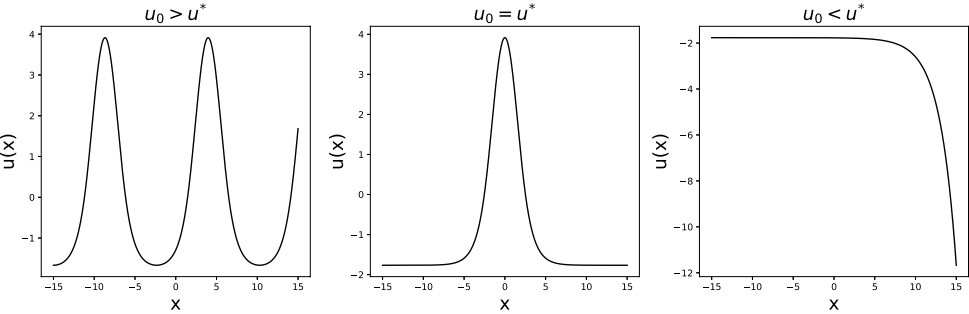

**Appendix 1—figure 2.** Solutions to *Equation 52* for $g = 1$, $w = -1.8$, $\gamma = 0$, $\Delta x = 0$. Then, keeping $u_0$ fixed, we can repeat a similar procedure to find $\Delta x$ for different values of $\gamma$. Also in this case, the solution either diverges or oscillates, apart from a single value $\Delta x^*$, for which the solution has the desired shape (see *Appendix 1—figure 3*). This eliminates the arbitrariness in the choice of $\Delta x$ since it imposes, for given $g$ and $w$, a relation $\Delta x = \Delta x^*(\gamma)$.

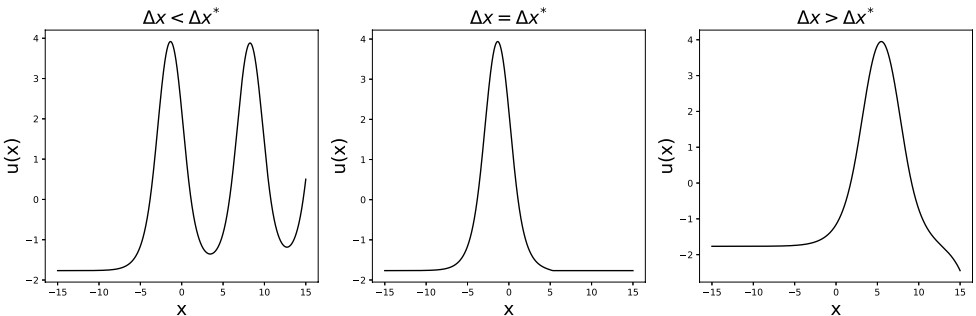

**Appendix 1—figure 3.** Solutions to *Equation 52* for $g = 1$, $w = -1.8$, $\gamma = 0.2$. We can then find the shape of the bump $u(x)$ for given values of $g$, $+w$ and $\gamma$, from which we can obtain the profile $v(x) = g\mathcal{N}(u(x))$ that we need for the calculation of the storage capacity. Some examples of the obtained profiles, for different values of $\gamma$, are shown in *Appendix 1—figure 4*.

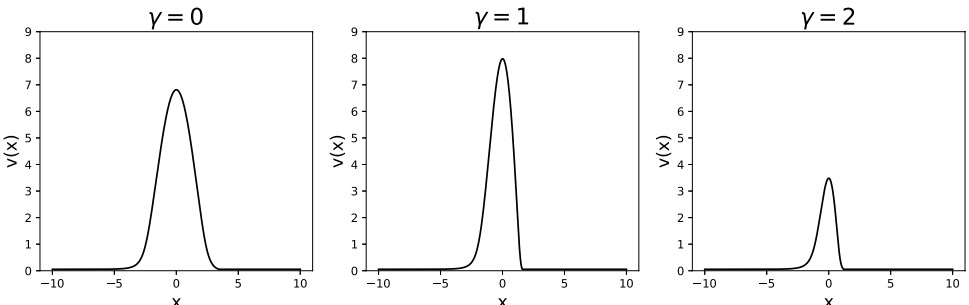

**Appendix 1—figure 4.** Activity profile $v(x)$, obtained for the same $g = 0.7$ and $w = -1.3$, at different values of $\gamma$. Plugging the obtained form of $v(x)$ into *Equation 29*, we can calculate the capacity. The dependence of the capacity on $\gamma$ is shown, for $L = 60$, in *Appendix 1—figure 5*.

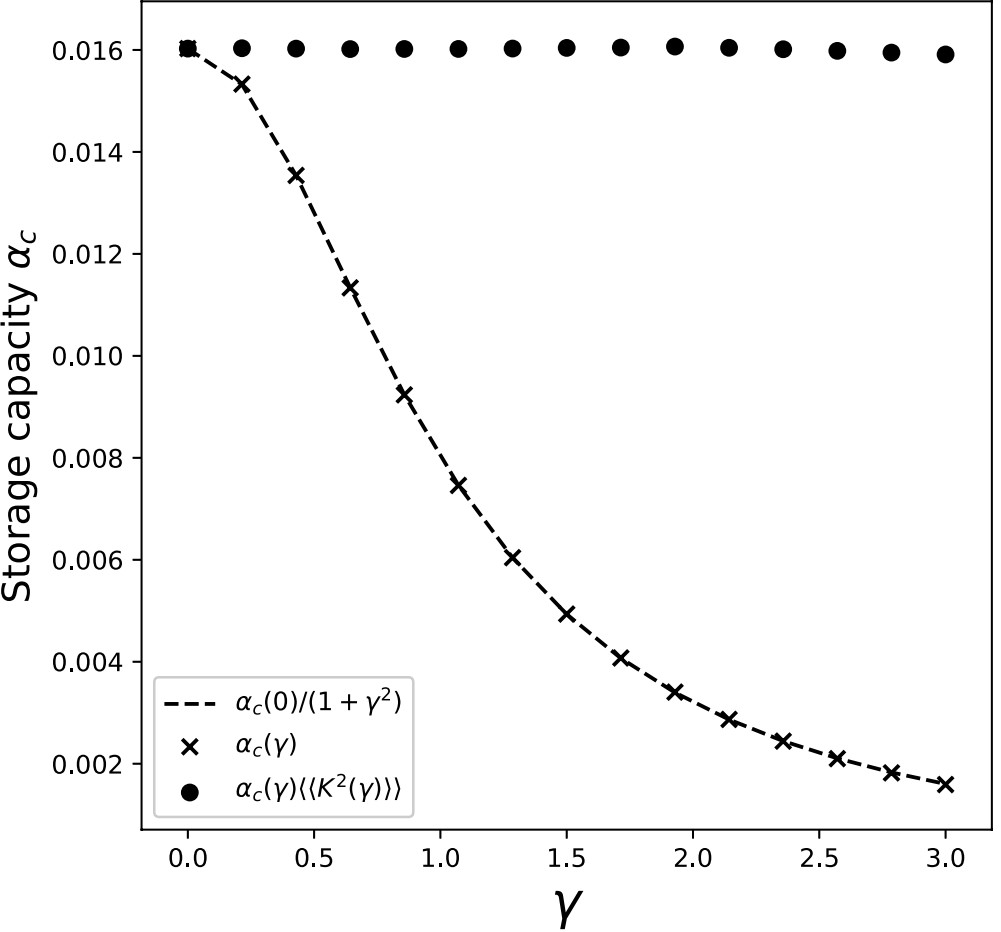

**Appendix 1—figure 5.** Dependence of the storage capacity on $\gamma$, for $L = 60$. The crosses show the full solution of *Equations 27 and 29*. The dashed line is obtained by taking the value of the capacity $\alpha(0)$ obtained with full solution at $\gamma = 0$, and multiplying it by the scaling of the kernel variance $(1 + \gamma^2)$. Full dots show the value of capacity obtained with the full solution and the contribution of the kernel variance factored out.

We can see from the full dots in the figure that the contribution of the integral in *Equation 29* is remarkably constant in $\gamma$. This is due to the fact that the distortions of the bump shape induced by the presence of the asymmetry have a negligible effect on the average square activity $y$, whose value is dominated by the dependence on $\gamma$ of the spatial variance of the kernel (*Equation 19*).

This allows us to approximate the value of the integral in *Equation 29* with its value in the $\gamma = 0$ case. We can then calculate the capacity as a function of $\gamma$ and $L$ by solving the symmetric case for different Ls, and then incorporating the dependence on $\gamma$ given by the kernel variance:

$$\alpha_c(L, \gamma) \sim \alpha_c(L, 0)/(1 + \gamma^2) \tag{53}$$

This approximation yields the results reported in the main text and in *Figure 8*

