## [Decision Letter]

**Acceptance summary:**

The study presents a new and elegant theoretical framework that generalizes memory storage and retrieval in neural networks to the dynamical case. Specifically, the authors show that imposing asymmetric interaction between units is sufficient to learn and retrieve sequential activation of these units, resembling the dynamics of hippocampal place cells during memory encoding and replay.

**Decision letter after peer review:**

[Editors’ note: the authors submitted for reconsideration following the decision after peer review. What follows is the decision letter after the first round of review.]

Thank you for submitting your work entitled "Continuous attractors for dynamic memories" for consideration by *eLife*. Your article has been reviewed by 2 peer reviewers, and the evaluation has been overseen by a Reviewing Editor and a Senior Editor. The following individual involved in review of your submission has agreed to reveal their identity: Ulisse Ferrari (Reviewer #3).

Our decision has been reached after consultation between the reviewers. Based on these discussions and the individual reviews below, we regret to inform you that this manuscript will not be considered further for publication in *eLife*. However, as pointed out by the reviewers, your work is of potential interest for the field. You may thus consider to submit a new version if you think you can significantly improve the manuscript. In this case, please include the original manuscript reference number as well as the names of the senior (Laura L Colgin) and reviewer editor (Adrien Peyrache) in your cover letter.

The authors present a computational model that allows for storing and reactivating dynamical memory patterns on low dimensional manifolds. The model presented in this manuscript is simple and elegant. Synaptic couplings are decomposed into a symmetric component, which makes low dimensional manifolds attractive, and an asymmetric component, which produces dynamic retrieval. This is of particular interest as sequential activation of neurons (i.e. trajectories on a manifold) is ubiquitous in the brain, especially in the hippocampus. This work is thus an important contribution to the field, bridging the gap between theoretical and experimental studies. However, the reviewers have raised several major concerns regarding the details of the model and its presentation. Furthermore, it seems necessary to demonstrate that the present model works well beyond the one-dimensional case and/or to test predictions of the model with actual experimental data.

*Reviewer #2:*

I really like the simplicity of the learning rule. However, the article lack clarity in many fundamental sections, and it took me really long to understand important details (if I understood them). When the authors introduce the model, they introduce the inhibition term, but then they remove it as they claim it can be reabsorbed in h_0_. From the equations this claim is not correct, assuming that h_0_ does not depend on the values of Vs. It's only by reading the methods that I understood that they actually choose dynamically both g and h_0_ to enforce a certain level of sparseness. I would say that already in the main text without introducing the term multiplying b at all, it is only confusing.

Even more confusing is the introduction of x. They define x as "preferential firing location in the stimulus space". They never spoke about stimuli, and I think they don't need to. I believe I finally understood what they mean, but I had to read the rest of the article (a couple of times). The description of the phenomenology is also unclear. In Figure 2 they never explain that they use x (y and z) to identify a neuron. So basically V is plotted as a function of the neuron number, parametrized by x. I guess that they discretized x and then assigned each x to a neuron. They should be more explicit about this, especially considering the broad audience of *eLife*. Moreover, they claim that the dynamics occurs in a low dimensional manifold, but in all the examples I see, their trajectory is always 1D, even in the cases in which they have multiple variables (e.g. figure 2c). Also in the dynamic kernel section they determine the kernel only in the case of one variable x, and obviously a 1D trajectory. The kernel they get is a for a rigid shift, under the assumption that the kernel is exponential. What would happen in a more general case? What would be the phenomenology if the dynamics really moves along a 2D or a 3D manifold? They only briefly mention in the Discussion that they consider only a single direction (they should say it from the very beginning). Also, what kind of asymmetric kernels do they consider? Is it only a rigid shift, or it could be a curvilinear trajectory when the underlying manifold has 2 or more dimensions? What is the maximal dimensionality of the trajectory? I guess it depends on the specific form of the asymmetric kernel. I have the impression that the authors have an interesting system to study, but they report only very few aspects of its dynamics (and in a rather confusing way).

Finally, they mention a few predictions in the Discussion, but they're rather generic. For a journal like *eLife*, in the absence of real data, quantitative predictions are important. The introduction, which is nicely written, contains several examples of phenomena that could be described by the proposed model.

There are several misprints and references missing.

*Reviewer #3:*

Spalla, Cornacchia and Treves presents a novel class of neural network models capable of storing and retrieving dynamical memory patterns. This result is a major advance in the field because it opens for many possible developments and importantly, it reduces the gap between theoretical and experimental works on memory. Previous models already showed the capacity of a neuronal network to store multiple memories, where each of them is represented by a pattern of neural activations. However, in these models when a particular memory is retrieved, neural activity aligns with the corresponding pattern, but it is static and does not flow along it. This issue prevents a clear connection between experimental and theoretical analysis of memory reactivations which are dynamical in nature, as for example in the case of the replay effect. Here instead, once neural activity aligns with a stored memory pattern, it evolves along it as a dynamical bump of activity. Therefore, here memories are dynamical objects, something closer to what has been observed in experiments.

Overall the paper presents some very relevant science and the claims are well supported by the presented analyses.

Strengths:

Having a simple but powerful model that allows for temporal evolution of reactivated memories represents a major step forward towards the construction neural network models with realistic phenomenology

Despite the added complexity of the present 'dynamical memory' model with respect to 'static memory' ones, storage capacity is not much affected, and can be even larger in certain regimes.

The authors' approach generalises previous static models, which are nicely recovered as limiting cases. This allows for encompassing previous results into a more flexible and powerful framework.

The behavior of the proposed model is robust against the fine details of its construction, meaning that a large class of models will present a similar phenomenology. This is important because it opens for considering more biologically plausible models that will still present the same phenomenological behavior.

Weaknesses:

Although the temporal evolution of the reactivated memory is the major advance of this work, their dynamics is very simple: once reactivated they evolve along a given (rigid) direction. It is not clear, for example, if the dynamics can be time-reversed, if cycles are allowed and if different memories can interact dynamically.

This work reduces the gap between computational and experimental studies of memory storing and retrieval. However it remains still very theoretical, and misses some development towards real neuronal networks. For example, it is suggested, but not shown that a biologically plausible mechanism of plasticity can result in systems that display the same phenomenology of the presented model. Additionally, the paper misses a clear discussion of how the model can empower the analysis of experimental results on memory storage and reactivation. Although it is fair to imagine that reaching these ambitious goals is a long term program that encompasses the current paper, the manuscript would benefit from a deep discussion of what are the main obstacles to be faced. As such, this weakness should be seen as suggestions for further developments, and not as limitation of the impact of the current work.

I find the dynamics of the retrieved memory either very simple, or not well explained and/or showcased. At first, it is not clear to me if the manifolds are compact and have endpoints. If it is the case, what happens when the dynamics reaches this point? Also, the velocity of the reactivation looks like a rigid property of the stored memory, but then the authors show that actually it might vary depending on the property of the systems (Figure 4, for example). Is there a way for actively tuning the velocity? This may link with the time-compressed reactivation observed in the replay? Additionally, it is not explained if the model can support more complex dynamical effects. For example, considering the effect of reverse replay (Foster and Wilson 2006), I wonder if the dynamics can be time-reversed, so that a dynamical memory can be reactivated in both directions. Also, is it possible to have cycles, so that the dynamics will run in a circle over and over? More generically, which kind of dynamics are supported by the current model? How to extend the model to increase the resulting phenomenology?

Additionally, I found the model's limitations discussed in LL 363->370 not clearly explained, and they should anyhow be expanded discussing what the model dynamics can or cannot support.

[Editors’ note: further revisions were suggested prior to acceptance, as described below.]

Thank you for resubmitting your work entitled "Continuous attractors for dynamic memories" for further consideration by *eLife*. Your revised article has been evaluated by Laura Colgin (Senior Editor) and a Reviewing Editor.

The manuscript has been greatly improved since the first submission. We believe that this manuscript is potentially of great interest for the community, however we think that the manuscript would benefit from one minor (but key) revision: to add a "real-life" example that would illustrate the potential of this theoretical work to a broad audience. Basically, this could be as simple as a toy model with place cells spiking along a linear/circular track. This would then entail to simulate the network spontaneous activity and measure the firing rate in time of a sub-sample of the cells (recorded cells), generate some spikes from the firing rate, show that the spiking activity of the network resembles that of replay events (apart from the reverse replay), and present the results in a way that an experimental reader will immediately understand the importance of the study.

We have also made some comments regarding the general presentation of the figures and equations, detailed below in the reviewer's specific comments. The authors can of course decide not to follow these suggestions as they are not critical for the main message of the paper.

*Reviewer #2:*

Spalla, Cornacchia and Treves have largely improved the manuscript and fully addressed all my main criticisms, and in my opinion also those of the other referee. I already found the paper interesting during the first review round, and now it has also improved in clarity and completeness. I particularly appreciated:

– The results of the paper are novel and very interesting. Even if the model is very simple and easy to understand, it presents an impressive phenomenology. Also, by just adding one more feature, it nicely extends previous models (the asymmetric component of the kernel), which are recovered when γ->0.

– The efforts that the authors have made to better characterise and describe the phenomenology of the dynamical retrieval: analyses as those of Figure 7 are very intriguing and in my opinion will open for further developments of the research field.

– The improvement of the exposure, which now better fits the scope and wide audience of *eLife*.

For all these reasons I recommend the publication of this paper.

*Reviewer #3:*

The study by Spalla et al. presents a new and elegant theoretical framework that generalizes memory storage and retrieval in auto associative networks to the dynamical case. Specifically, while purely symmetrical synapses (as in canonical attractor networks) lead to the retrieval of static patterns, asymmetric synapses lead to sequential activation of units in the network, as observed for example during replay of hippocampal place cells. In addition, the study demonstrates that, as previous models had shown in the static case, such network can store multiple maps.

The main concerns with this study, in its present form, is the general presentation of the model and of its general properties, which could be hard to follow for a non-expert reader. The study would also benefit from concrete examples, such as simulation of replay in a minimal model of hippocampal place cells.

The study by Spalla et al. presents a new and elegant theoretical framework that generalizes memory storage and retrieval in auto associative networks to the dynamical case. Specifically, asymmetric synapses lead to sequential activation of units in the network, instead of the classical static patterns emerging from purely symmetrical synapses (e.g. attractors in which connectivity kernels depends only on the "representational distance" between units in the feature space). The study also demonstrates that, as previous models had shown in the static case, such network can also store multiple maps.

While the authors have already addressed many comments made during a previous round of review, the general presentation of the manuscript should be greatly improved for publication in a life science journal targeting a broad audience. Importantly, the number of main figures should be drastically reduced, and figures should focus only on the most important claims of the study.

One major concern though is that the study would benefit from a concrete example. For example, displaying simulated "replayed" trajectories in 2D maps (as a neurophysiologist would present it in an experimental study) in different cases influencing the speed (and general behaviour) of replay, such as different levels of asymmetry and sparsity.

The remarks below are related to the specific presentation of the manuscript.

Top panels of Figure 3 and associated text should be put in supplementary info, as this section can be fairly confusing for the non-expert reader and does not contribute much to the understanding of the phenomenon.

Figure 4 conveys an important message: the speed of the bump is related to the asymmetry of the connectivity kernel. The figure legend should be clarified: x-axis should read "kernel asymmetry (\γ)", with arrows point to either left ("more symmetrical") to right ("more asymmetrical"). |dx| should be replace by "bump speed (a.u.)". The message of Figure 4b is unclear and should certainly be discarded.

While Figure 5 shows that the solution can be generalized to a broad class of asymmetric kernels, Figure 6 presents again the link between network parameters and bump features (speed, etc.) in the case of an exponential kernel. One idea would be to merge the bottom panels of Figure 3, Figure 4a and Figure 6 to convey one message about the exponential case. Then show (current) Figure 5 to show that it is valid in other cases.

Figure 7 presents an interesting simulation, illustrating that retrieval can be specific to a particular map. Here again, the presentation should be drastically improved: x_1,2,3 should be replaced by more explicit terms like "feature space (map 1)" or something similar. Y-axis should be labels "activity level (V)" or something similar, etc. Against, it is important to present figure that do not require the reader to rely too much on abbreviations and variable names.

The link between kernel asymmetry and storage capacity is important but should be made clearer. Most of the equations (17-29) should be put in appendix and be limited to equation 30 and 31. Figure 8 and 9 present overlapping data, they should be merged (or better: only one of the two should be shown) and the same remarks as for Figure 4 apply for the presentation of this figure.

The presentation of the non-monotonic dependence of storage capacity w.r.t \γ in the fully connected case is another aspect that should be improved. Figure 10 would benefit from an additional graph showing P=50% chance retrieval as a function of γ to illustrate the non-monotonic relationship. This figure should be merge in a multi-panel figure with Figure 11 so that the reader can immediately grasp the main messages regarding the fully connected case.

---

## [Author Response]

[Editors’ note: the authors resubmitted a revised version of the paper for consideration. What follows is the authors’ response to the first round of review.]

Reviewer #2:I really like the simplicity of the learning rule. However, the article lack clarity in many fundamental sections, and it took me really long to understand important details (if I understood them). When the authors introduce the model, they introduce the inhibition term, but then they remove it as they claim it can be reabsorbed in h_0_. From the equations this claim is not correct, assuming that h_0_ does not depend on the values of Vs. It's only by reading the methods that I understood that they actually choose dynamically both g and h_0_ to enforce a certain level of sparseness. I would say that already in the main text without introducing the term multiplying b at all, it is only confusing.

We have clarified in the main text (LL 120-126) how the inhibition mechanism works, and simplified the notation, avoiding the complication of the introduction of the b(x) term.

Even more confusing is the introduction of x. They define x as "preferential firing location in the stimulus space". They never spoke about stimuli, and I think they don't need to. I believe I finally understood what they mean, but I had to read the rest of the article (a couple of times).

We have expanded the section “A mechanistic model for dynamic retrieval” (LL 127-139), clarifying the meaning of the parameter x and avoiding references to stimuli, and striving for a clearer and more accessible explanation of the model.

The description of the phenomenology is also unclear. In Figure 2 they never explain that they use x (y and z) to identify a neuron. So basically V is plotted as a function of the neuron number, parametrized by x. I guess that they discretized x and then assigned each x to a neuron. They should be more explicit about this, especially considering the broad audience of eLife.

We have clarified the phenomenology presented in Figure 2, by rewriting and expanding the description of the plots in the main text (LL 174-183)

Moreover, they claim that the dynamics occurs in a low dimensional manifold, but in all the examples I see, their trajectory is always 1D, even in the cases in which they have multiple variables (e.g. figure 2c). Also in the dynamic kernel section they determine the kernel only in the case of one variable x, and obviously a 1D trajectory.

We clarify this aspect in LL 207-212 and in the discussion (LL 500-506), underlying how the dynamics in 2D and 3D is different from a simple 1D trajectory.

We have also added a new analysis of the case of two intersecting 1D trajectories embedded in a 2D manifold (Figure 2(e) and accompanying text), which explores the case of position-dependent asymmetry, and provides a proof of concept of the possibility to expand the model in this direction.

The kernel they get is a for a rigid shift, under the assumption that the kernel is exponential.

This problem is addressed in the section “dynamical retrieval is robust” (LL 266-283). There we investigate the effect of the kernel shape and its parameters on the behaviour of the model. We find this behaviour to be largely independent from the specific kernel shape, and able to produce dynamic retrieval in a remarkably broad range of parameters (see Figure 4 and Figure 5).

What would happen in a more general case? What would be the phenomenology if the dynamics really moves along a 2D or a 3D manifold? They only briefly mention in the Discussion that they consider only a single direction (they should say it from the very beginning). Also, what kind of asymmetric kernels do they consider? Is it only a rigid shift, or it could be a curvilinear trajectory when the underlying manifold has 2 or more dimensions? What is the maximal dimensionality of the trajectory? I guess it depends on the specific form of the asymmetric kernel. I have the impression that the authors have an interesting system to study, but they report only very few aspects of its dynamics (and in a rather confusing way).

We have added clarifications on these aspects as stated in the previous points.

In the case with a constant asymmetric direction, the presence of the asymmetry imposes no additional bound on the maximal dimensionality of the manifold.

As it is the case in continuous attractor networks of any kind, a trade-off is to be expected between the resolution and the dimensionality of the representation, if the size of the network is kept fixed.

Finally, they mention a few predictions in the Discussion, but they're rather generic. For a journal like eLife, in the absence of real data, quantitative predictions are important. The introduction, which is nicely written, contains several examples of phenomena that could be described by the proposed model.

We have added two key quantitative predictions by the model: the predicted elongation of place fields in the direction of the running trajectory (LL 224-227, and LL 506-513) and an interaction between the sparsity of the activity of the network and the retrieval speed (LL 191- 196 and LL 452-459).

These predictions complement the main result of the paper: that auto associative memory networks can encode continuous dynamic memory with large capacity, and that the asymmetry of connections, often considered as an afterthought in mechanistic models, can be a crucial ingredient of memory systems.

We believe these points are of interest to the broad *eLife* audience, beyond the technical apparatus developed to obtain them.

There are several misprints and references missing.

We have made an effort to correct all misprints and missing references, and we have added several more references to better put the work in the context of the existing literature.

Reviewer #3:Spalla, Cornacchia and Treves presents a novel class of neural network models capable of storing and retrieving dynamical memory patterns. This result is a major advance in the field because it opens for many possible developments and importantly, it reduces the gap between theoretical and experimental works on memory. Previous models already showed the capacity of a neuronal network to store multiple memories, where each of them is represented by a pattern of neural activations. However, in these models when a particular memory is retrieved, neural activity aligns with the corresponding pattern, but it is static and does not flow along it. This issue prevents a clear connection between experimental and theoretical analysis of memory reactivations which are dynamical in nature, as for example in the case of the replay effect. Here instead, once neural activity aligns with a stored memory pattern, it evolves along it as a dynamical bump of activity. Therefore, here memories are dynamical objects, something closer to what has been observed in experiments.Overall the paper presents some very relevant science and the claims are well supported by the presented analyses.Strengths:Having a simple but powerful model that allows for temporal evolution of reactivated memories represents a major step forward towards the construction neural network models with realistic phenomenologyDespite the added complexity of the present 'dynamical memory' model with respect to 'static memory' ones, storage capacity is not much affected, and can be even larger in certain regimes.The authors' approach generalises previous static models, which are nicely recovered as limiting cases. This allows for encompassing previous results into a more flexible and powerful framework.The behavior of the proposed model is robust against the fine details of its construction, meaning that a large class of models will present a similar phenomenology. This is important because it opens for considering more biologically plausible models that will still present the same phenomenological behavior.Weaknesses:Although the temporal evolution of the reactivated memory is the major advance of this work, their dynamics is very simple: once reactivated they evolve along a given (rigid) direction. It is not clear, for example, if the dynamics can be time-reversed, if cycles are allowed and if different memories can interact dynamically.This work reduces the gap between computational and experimental studies of memory storing and retrieval. However it remains still very theoretical, and misses some development towards real neuronal networks. For example, it is suggested, but not shown that a biologically plausible mechanism of plasticity can result in systems that display the same phenomenology of the presented model. Additionally, the paper misses a clear discussion of how the model can empower the analysis of experimental results on memory storage and reactivation. Although it is fair to imagine that reaching these ambitious goals is a long term program that encompasses the current paper, the manuscript would benefit from a deep discussion of what are the main obstacles to be faced. As such, this weakness should be seen as suggestions for further developments, and not as limitation of the impact of the current work.I find the dynamics of the retrieved memory either very simple, or not well explained and/or showcased. At first, it is not clear to me if the manifolds are compact and have endpoints. If it is the case, what happens when the dynamics reaches this point?

We have clarified in the presentation of the model (LL 158-165) that we consider manifolds with periodic boundary conditions, in which the dynamic retrieval produces periodic cycles. We highlight that this simplifying assumption is not a necessary feature of the model, and we study the alternative case in which this is replaced by a link between the endpoint of a manifold and the beginning of the next one (Figure 7 (c), (d) and accompanying text.). This analysis not only shows that dynamic retrieval does not rely on the specific boundary conditions, but also demonstrates that a chain of multiple continuous manifolds can be retrieved sequentially by the network.

Also, the velocity of the reactivation looks like a rigid property of the stored memory, but then the authors show that actually it might vary depending on the property of the systems (Figure 4, for example). Is there a way for actively tuning the velocity? This may link with the time-compressed reactivation observed in the replay?

We have further expanded the study of the dependence of the retrieval speed on the parameters of the network by adding an analysis of the effect of sparsity on the reactivation velocity (Figure 2 (d) and Figure 3 (b), (c)).

We find that the sparsity of the representation (a feature of the network activity, that can be regulated instantaneously during the dynamics) interacts with the asymmetry parameter \γ in determining the retrieval speed.

The possibility of instantaneously tuning the velocity by acting on the sparsity of the activity also yields a key new prediction: in a network with fixed connectivity, retrieval speed can be modulated by the activity level of the population.

Additionally, it is not explained if the model can support more complex dynamical effects. For example, considering the effect of reverse replay (Foster and Wilson 2006), I wonder if the dynamics can be time-reversed, so that a dynamical memory can be reactivated in both directions. Also, is it possible to have cycles, so that the dynamics will run in a circle over and over? More generically, which kind of dynamics are supported by the current model? How to extend the model to increase the resulting phenomenology?

We have added the clarification that the current model cannot handle backward and forward replay of the same trajectory (LL 196-198 and LL 514-521 in the discussion). The dynamics is naturally organized in cycles under the assumption of periodic boundary conditions, but this is not the only possibility the model can describe.

To clarify this point, we have added new analyses that demonstrate complex dynamical effects. In particular the linking of dynamical memories and the possibility to simultaneously memorize manifolds with different values of the asymmetric component, leading to different retrieval speeds (Figure 7 and accompanying text).

Additionally, I found the model's limitations discussed in LL 363->370 not clearly explained, and they should anyhow be expanded discussing what the model dynamics can or cannot support.

As stated in response to reviewer 2: we clarify this aspect in LL 207-212 and in the discussion (LL 500-506), underlying how the dynamics in 2D and 3D is different from a simple 1D trajectory.

We have also added a new analysis of the case of two intersection 1D trajectories embedded in a 2D manifold (Figure 2(e) and accompanying text), which explores the case of position-dependent asymmetry, and provides a proof of concept of the possibility to expand the model in this direction.

[Editors’ note: what follows is the authors’ response to the second round of review.]

Reviewer #3:The study by Spalla et al. presents a new and elegant theoretical framework that generalizes memory storage and retrieval in auto associative networks to the dynamical case. Specifically, while purely symmetrical synapses (as in canonical attractor networks) lead to the retrieval of static patterns, asymmetric synapses lead to sequential activation of units in the network, as observed for example during replay of hippocampal place cells. In addition, the study demonstrates that, as previous models had shown in the static case, such network can store multiple maps.The main concerns with this study, in its present form, is the general presentation of the model and of its general properties, which could be hard to follow for a non-expert reader. The study would also benefit from concrete examples, such as simulation of replay in a minimal model of hippocampal place cells.The study by Spalla et al. presents a new and elegant theoretical framework that generalizes memory storage and retrieval in auto associative networks to the dynamical case. Specifically, asymmetric synapses lead to sequential activation of units in the network, instead of the classical static patterns emerging from purely symmetrical synapses (e.g. attractors in which connectivity kernels depends only on the "representational distance" between units in the feature space). The study also demonstrates that, as previous models had shown in the static case, such network can also store multiple maps.While the authors have already addressed many comments made during a previous round of review, the general presentation of the manuscript should be greatly improved for publication in a life science journal targeting a broad audience. Importantly, the number of main figures should be drastically reduced, and figures should focus only on the most important claims of the study.

We thank the reviewer for the suggestion. We have reduced the number of main figures by merging the last three figures in what is now Figure 8, and have strived to improve graphical quality and figure accessibility by a general standardization of figure design and a relabeling of figure axes throughout the manuscript, with the aim to make them more consistent and more readily interpretable.

One major concern though is that the study would benefit from a concrete example. For example, displaying simulated "replayed" trajectories in 2D maps (as a neurophysiologist would present it in an experimental study) in different cases influencing the speed (and general behaviour) of replay, such as different levels of asymmetry and sparsity.

We agree with the reviewer that a concrete example is beneficial for the interpretation of the study. To this end, we have added the analysis of a simulated electrophysiological experiment probing the population activity during dynamic retrieval (see also response to the editors).

This simulation is illustrated in Figure 6(c), constructed so as to be familiar to experimental readers in its design and interpretation.

The figure and the accompanying text (LL 314-328, plus appendix B for details) serve the double purpose to connect the behaviour of the model to experimentally observable quantities, and to further illustrate the concept of dynamic auto associative memory, that Figure 6 was originally built to exemplify.

The remarks below are related to the specific presentation of the manuscript.Top panels of Figure 3 and associated text should be put in supplementary info, as this section can be fairly confusing for the non-expert reader and does not contribute much to the understanding of the phenomenon.

We believe that the analytical solvability of the model is a key feature of our work that can yield quantitative insights and a deep understanding of the mechanisms behind dynamic retrieval. Therefore, we chose to maintain Figure 3 and the accompanying paragraph in the main text. However, during the first revision we moved all technical details to Appendix C, including the previous Figure 3 illustrating the analytical solution procedure, leaving only results and interpretation in the main text.

We have tried to improve the graphical design and axis labeling to make the figure more accessible to a non-technical audience.

Figure 4 conveys an important message: the speed of the bump is related to the asymmetry of the connectivity kernel. The figure legend should be clarified: x-axis should read "kernel asymmetry (\γ)", with arrows point to either left ("more symmetrical") to right ("more asymmetrical"). |dx| should be replace by "bump speed (a.u.)". The message of Figure 4b is unclear and should certainly be discarded.

We think the reviewer is referring to the version of Figure 4 that was in the original manuscript. We indeed changed this figure during our first revision, by getting rid of Figure 4b, as the reviewer suggested, and incorporating Figure 4a, with an extended analysis on the effect of sparsity, in what is now Figure 3. We have also changed axis label names to improve clarity.

While Figure 5 shows that the solution can be generalized to a broad class of asymmetric kernels, Figure 6 presents again the link between network parameters and bump features (speed, etc.) in the case of an exponential kernel. One idea would be to merge the bottom panels of Figure 3, Figure 4a and Figure 6 to convey one message about the exponential case. Then show (current) Figure 5 to show that it is valid in other cases.

We agree with the reviewer and thank them for the suggestion. We swapped the two figures (they are now Figure 4, on the effects of the parameters in the exponential case and Figure 5, on different kernel shapes) and restructured the section on robustness of the asymmetric interactions by examining in depth the exponential kernel case first, and then extending the analysis to other kernel shapes.

Figure 7 presents an interesting simulation, illustrating that retrieval can be specific to a particular map. Here again, the presentation should be drastically improved: x_1,2,3 should be replaced by more explicit terms like "feature space (map 1)" or something similar. Y-axis should be labels "activity level (V)" or something similar, etc. Against, it is important to present figure that do not require the reader to rely too much on abbreviations and variable names.

We have changed the labels in this figure, as well as in the others throughout the manuscript, following this advice for which we are thankful to the reviewer. We have also made some changes in coloring and label sizes to improve readability, as well as adding here as a new panel the results of the new simulation of experimental recordings (see main points).

The link between kernel asymmetry and storage capacity is important but should be made clearer. Most of the equations (17-29) should be put in appendix and be limited to equation 30 and 31. Figure 8 and 9 present overlapping data, they should be merged (or better: only one of the two should be shown) and the same remarks as for Figure 4 apply for the presentation of this figure.

We believe that the reviewer is here referring to the first version of the manuscript.

During our first revision we heavily restructured this section putting most of the mathematical details to Appendix F. Also Figure 8 was moved to the appendix, since we agree with the reviewer that the data is overlapping with Figure 9 (now Figure 8a), and the point of the figure (to show the goodness of our approximation ansatz) is beyond the scope of the main text. Following the previous point on the number of main figures, we have aggregated all figures of this section in what is now Figure 8, that conveys in a single place the main messages on the storage capacity of the network.

The presentation of the non-monotonic dependence of storage capacity w.r.t \γ in the fully connected case is another aspect that should be improved. Figure 10 would benefit from an additional graph showing P=50% chance retrieval as a function of γ to illustrate the non-monotonic relationship. This figure should be merge in a multi-panel figure with Figure 11 so that the reader can immediately grasp the main messages regarding the fully connected case.

We have merged the figures in what is now Figure 8. We did not insert an additional graph, with the rationale not to show overlapping data and not to multiply further the number of figures. However, we have changed labels and other graphical aspects to improve understandability and readability. We believe that the non-monotonicity of the storage capacity is appreciable by the reader from both Figure 8b, where the transition point is shown to move back and forth with γ, and Figure 8c, in which the joint dependence of the capacity on γ and 1/L shows a clear peak in the bulk of the parameter ranges.